# CARD14$^{E138A}$ signalling in keratinocytes induces TNF-dependent skin and systemic inflammation

Joan Manils[1,2†], Louise V Webb[1†], Ashleigh Howes[3], Julia Janzen[2], Stefan Boeing[1,4,5], Anne M Bowcock[3,6*], Steven C Ley[2*]

[1]The Francis Crick Institute, London, United Kingdom; [2]Department of Immunology & Inflammation, Imperial College London, London, United Kingdom; [3]National Heart & Lung Institute, Imperial College London, London, United Kingdom; [4]Bioinformatics and Biostatistics, The Francis Crick Institute, London, United Kingdom; [5]Crick Scientific Computing - Digital Development Team, The Francis Crick Institute, London, United Kingdom; [6]Departments of Oncological Science, Dermatology, and Genetics & Genome Sciences, Icahn School of Medicine at Mount Sinai, New York, United States

**\*For correspondence:**
anne.bowcock@mssm.edu (AMB);
sley@imperial.ac.uk (SCL)

[†]These authors contributed equally to this work

**Competing interests:** The authors declare that no competing interests exist.

**Abstract** To investigate how the *CARD14*$^{E138A}$ psoriasis-associated mutation induces skin inflammation, a knock-in mouse strain was generated that allows tamoxifen-induced expression of the homologous *Card14*$^{E138A}$ mutation from the endogenous mouse *Card14* locus. Heterozygous expression of CARD14$^{E138A}$ rapidly induced skin acanthosis, immune cell infiltration and expression of psoriasis-associated pro-inflammatory genes. Homozygous expression of CARD14$^{E138A}$ induced more extensive skin inflammation and a severe systemic disease involving infiltration of myeloid cells in multiple organs, temperature reduction, weight loss and organ failure. This severe phenotype resembled acute exacerbations of generalised pustular psoriasis (GPP), a rare form of psoriasis that can be caused by *CARD14* mutations in patients. CARD14$^{E138A}$-induced skin inflammation and systemic disease were independent of adaptive immune cells, ameliorated by blocking TNF and induced by CARD14$^{E138A}$ signalling only in keratinocytes. These results suggest that anti-inflammatory therapies specifically targeting keratinocytes, rather than systemic biologicals, might be effective for GPP treatment early in disease progression.

## Introduction

Psoriasis is a chronic auto-immune skin disease that affects 2–3% of the European population. Disease characteristics include epidermal hyperplasia, infiltration of leukocytes in the dermis and erythema (*Dainichi et al., 2018*). Long-term activation of the immune system in the skin can also lead to tissue damage in other organs, and psoriasis is generally regarded as a systemic disease. Consequently, numerous comorbidities, including psoriatic arthritis, inflammatory bowel disease (IBD) and cardiovascular disease, are associated with psoriasis (*Greb et al., 2016*).

Genome-wide association studies have revealed over 60 loci that contribute to the development of psoriasis. These include candidate genes involved in skin barrier function and the immune response (*Liang et al., 2017a*). It is thought that many low risk genetic variants together with environmental factors increase the predisposition of individuals to psoriasis development. Psoriasis vulgaris (PV), the most common form, is characterised by stable erythematous scaly plaques, predominantly located on knees and elbows, in which the adaptive immune system predominates. In contrast, generalised pustular psoriasis (GPP), a very rare form of the disease that is characterised by widespread diffuse skin inflammation and sub-corneal pustules (Munro's microabscesses and

spongiform pustules of Kogoj), involves massive infiltration of neutrophils in the skin. GPP patients also experience acute exacerbations (flare ups) involving development of new skin lesions which can cover >70% of the body surface, fever, cheilitis (scaling and superficial ulceration of the lips), diarrhoea and dehydration (*Ly et al., 2019*). GPP exacerbations, which often meet the criteria for systemic inflammatory response syndrome, involve neutrophil infiltration in multiple organs, are difficult to treat and associated with significant morbidity and mortality (3–7%) (*Gooderham et al., 2019*).

Gain-of-function, highly penetrant (>90%) and dominantly acting mutations within the *CARD14* gene can trigger the development of PV or GPP (*Jordan et al., 2012b*). CARD14 (CARMA2) is a member of the CARMA family of scaffolding proteins that includes CARD11 (CARMA1) and CARD10 (CARMA3) (*Lu et al., 2019*). Each of these proteins has a similar domain structure, comprising an N-terminal CARD domain, followed by a coiled-coil (CC) domain, and a C-terminal MAGUK domain (PDZ-SH3-GUK). CARD11 and CARD10 play critical roles in the activation of NF-κB transcription factors following ligation of antigen receptors and G-protein-coupled receptors, respectively (*Lu et al., 2019*). NF-κB, composed of dimers of Rel polypeptides, regulates gene expression by binding to κB elements in the promoters and enhancers of multiple target genes that control immune and inflammatory responses (*Zhang et al., 2017*).

The structural similarity to CARD11 and CARD10 suggests a role for CARD14 in NF-κB activation. Consistent with this, the highly penetrant $CARD14^{E138A}$ psoriasis-associated mutation, first identified as a sporadic mutation in a child suffering from GPP (*Jordan et al., 2012b*), induces CARD14 interaction with BCL10 (B cell lymphoma protein 10) and the paracaspase MALT1 (mucosa-associated lymphoid tissue lymphoma translocation protein 1) (*Howes et al., 2016*). This complex triggers activation of the IκB kinase (IKK) complex, the central activator of NF-κB transcription factors, leading to constitutive activation of NF-κB and expression of pro-inflammatory cytokine/chemokine genes in keratinocytes in the absence of exogenous stimulation.

Recent studies have investigated how psoriasis-associated *CARD14* mutations induce skin inflammation by generating knock-in mice expressing the mouse equivalent CARD14 variants (*Mellett et al., 2018*; *Sundberg et al., 2019*; *Wang et al., 2018*). These mice develop psoriasiform skin inflammation that is partially dependent on the cytokines IL-17A and IL-23, which play important roles in human psoriasis (*Greb et al., 2016*). Although these studies have confirmed the importance of CARD14 mutations in inducing skin inflammation, the constitutive nature of the knock-in mutations generated have precluded detailed study of disease pathogenesis. Furthermore, the constitutive $CARD14^{E138A}$ mutation is associated with embryonic or peri-natal lethality, preventing analysis of disease in adult skin. To circumvent these limitations, we generated a new knock-in mouse allele $Card14^{LSL-E138A}$ that allows conditional expression of the human $CARD14^{E138A}$ mutation in the homologous mouse locus following tamoxifen injection. The inducible nature of this mutation allowed us to study the direct effects of $CARD14^{E138A}$ signalling, rather than potentially indirect secondary or tertiary consequences of constitutively expressed $CARD14^{E138A}$. Tamoxifen-induced expression of $Card14^{E138A}$ in adult mice also ruled out any developmental effects of $CARD14^{E138A}$ signalling on skin inflammation.

We present evidence that conditional heterozygous expression of $CARD14^{E138A}$ in adult mice rapidly promoted psoriasiform skin inflammation, which was independent of adaptive immune cells and due to signalling in keratinocytes. However, homozygous expression of $CARD14^{E138A}$ induced rapid weight loss, temperature reduction and extensive neutrophil infiltration in multiple organs. This severe systemic phenotype resulted from $CARD14^{E138A}$ signalling in keratinocytes and showed striking similarities to an acute flare of GPP in human patients. Heterozygous and homozygous phenotypes were both ameliorated by antibody neutralisation of tumour necrosis factor (TNF). Our findings suggest that specific blockade of the CARD14 signalling pathway in keratinocytes would inhibit the systemic inflammatory syndrome that can develop in GPP patients with activating CARD14 mutations.

## Results

Human and mouse CARD14 proteins are highly homologous, with 77% amino acid identity overall and 80% identity in the coiled-coil region in which the $CARD14^{E138A}$ mutation is located (*Figure 1— figure supplement 1A*). To study how $CARD14^{E138A}$ mutation induces skin inflammation, we developed a conditional knock-in mouse strain, $Card14^{LSL-E138A/LSL-E138A}$, in which a floxed minigene

(exons 4–22) encoding wild type (WT) C-terminus of CARD14 linked to a 3xFLAG tag was inserted between exons 3 and 4 of the *Card14* locus and an E138A point mutation was introduced into endogenous *Card14* exon 5 (*Figure 1—figure supplement 1B*). In the absence of Cre-mediated recombination, *Card14* was expressed from exon 3 and the inserted minigene to produce WT CARD14-3xFLAG. After Cre-mediated recombination, the minigene was excised, allowing transcription of *Card14* from the endogenous exons and expression of CARD14$^{E138A}$.

## CARD14 is expressed at high levels in differentiated keratinocytes of the skin epidermis

In order to understand the effects of *Card14*$^{E138A}$ mutation, it was first important to know where CARD14 protein is normally expressed. High levels of CARD14-3XFLAG were detected in the back skin and ears, as expected, and also in the colon, small intestine and caecum (*Figure 1A*). Low levels of CARD14-3XFLAG were additionally found in the liver and kidney.

RNAscope analysis revealed that *Card14* mRNA expression in healthy mouse skin was restricted to the upper (most differentiated) keratinocytes, the outer root sheath of the hair and sebocytes in the skin (*Figure 1B*). These data contrasted with earlier immunohistochemistry results suggesting CARD14 expression in the basal layer of human skin (*Fuchs-Telem et al., 2012*; *Jordan et al., 2012b*). However, the RNAscope analyses were in line with published RNA sequencing data from different layers of the mouse epidermis (*Figure 1—figure supplement 1C*) and single cell sequencing of murine epidermis (http://kasperlab.org/mouseskin) which also indicated *Card14* mRNA expression in differentiated keratinocytes (*Asare et al., 2017*). Consistent with this, CARD14 protein expression was induced in mouse keratinocytes following in vitro differentiation in culture medium containing CaCl$_2$ (*Figure 1—figure supplement 1D*; *Bikle et al., 2012*). *CARD14* mRNA and CARD14 protein expression was similarly induced in cultured human keratinocytes concurrent with the keratinocyte differentiation marker involucrin (*Figure 1—figure supplement 1E and F*). The *CARD14* mRNA expression data agree with published transcriptomic datasets of differentiating human keratinocytes (*Bin et al., 2016*) and organotypic epidermis (*Lopez-Pajares et al., 2015*; *Figure 1—figure supplement 1G*).

## Heterozygous expression of CARD14$^{E138A}$ rapidly induces psoriasiform skin inflammation

Psoriasis-associated *CARD14* mutations in humans are generally heterozygous (*Jordan et al., 2012a*). *Card14*$^{LSL-E138A/+}$*Rosa26*$^{CreERT2}$ mice were generated to test the effect of ubiquitous heterozygous expression of *Card14*$^{E138A}$ following tamoxifen-induced minigene deletion (*Seibler et al., 2003*), according to the scheme in *Figure 1C*.

A time course experiment revealed that intraperitoneal tamoxifen promoted epidermal thickening (acanthosis), hyperkeratosis and keratinocyte proliferation (as indicated by Ki67 staining) in the ears 4–5 days (d) after the first injection (*Figure 1D*). Staining for the Ca$^{2+}$-binding protein S100A9 showed CARD14$^{E138A}$ increased infiltration of S100A9$^+$ myeloid cells and also enhanced S100A9 expression in keratinocytes, which express S100A9 only following cell stress (*Eckert et al., 2004*). CARD14$^{E138A}$ induced a broader distribution of the keratinocyte differentiation marker involucrin in the epidermis, indicative of altered keratinocyte differentiation. Increased staining for endomucin, an endothelial marker, showed that CARD14$^{E138A}$ promoted neoangiogenesis. All these changes were also detected in the back skin of the mice following tamoxifen induction (*Figure 1—figure supplement 2A*). In addition, mRNA expression of the psoriasis-associated genes *Il17a, Tnfa, S100a9, Il17c* and *Il1f9* was induced by tamoxifen in the ears and the back skin (*Figure 1E* and *Figure 1—figure supplement 2B*).

Flow cytometry further characterised immune cells in the ears five days (5d) after tamoxifen injection (for gating strategy see *Figure 1—figure supplement 3*). There was an extensive infiltration of immune cells in the ears of the *Card14*$^{LSL-E138A/+}$*Rosa26*$^{CreERT2}$ mice compared to *Card14*$^{+/+}$*Rosa26*$^{CreERT2}$ controls. Neutrophils, macrophages, dendritic cells (DC), CD4$^+$ T cells, CD8$^+$ T cells, regulatory T cells (Tregs), γδ T cells and NK cells were all found in significantly increased numbers 5d after tamoxifen injection (*Figure 1F*).

Together these results indicated that expression of CARD14$^{E138A}$ promoted the rapid development of skin inflammation that shared many features in common with psoriasis (*Greb et al., 2016*).

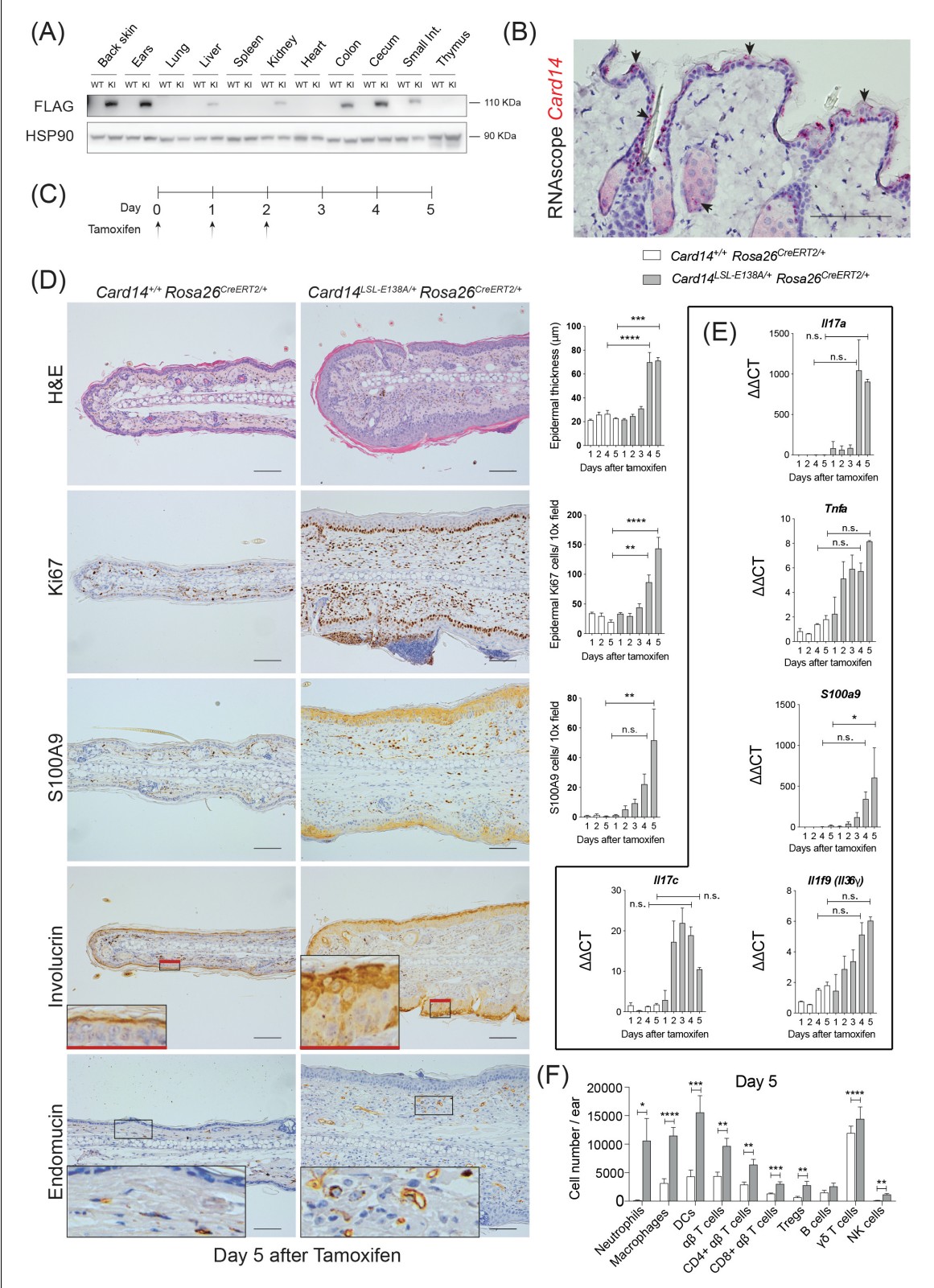

**Figure 1.** Conditional expression of CARD14[E138A] in *Card14*[LSL-E138A/+] *Rosa26*[CreERT2/+] mice induces rapid changes in ear skin histology. (A) CARD14-FLAG protein expression analysed in total extracts from different organs from control (WT) and *Card14*[E138A-LSL] (KI) mice by immunoblotting. (B) Localisation of *Card14* mRNA expression in the skin assessed by RNAscope. (C) Timeline of the *Card14*[E138A] induction by tamoxifen and sample collection. (D) Representative histology images of ears on d5 after tamoxifen injection: (first panel) H and E staining and acanthosis quantification over
*Figure 1 continued on next page*

Figure 1 continued

time; (second panel) Ki67 staining at d5 and quantitation; (third panel) S100A9 staining at d5 and quantitation; (fourth panel) involucrin staining at d5 and (bottom panel) endomucin staining at d5. (E) qRT-PCR analysis of the expression of *IL17a, Tnfa, S100a9, Il17c* and *Il1f9* mRNAs. (F) Quantification and characterisation of the immune cell infiltrate of the ears at d5 after tamoxifen by FACS. Data pooled from 4 independent experiments; *Card14*$^{+/+}$ *Rosa26*$^{CreERT2/+}$ n = 22, *Card14*$^{LSL-E138A/+}$ *Rosa26*$^{CreERT2/+}$ 5d n = 22. Data collected from a mixture of male and female mice. (B, D) Scale bar = 100 µm. Differences between groups analysed by one-way ANOVA (D and E) or Student's t-test (F). *, p<0.05; **, p<0.01; ***, p<0.001; ****, p<0.0001. For clarity, only statistical analyses between the two genotypes at day 4 and day 5 were noted.

The online version of this article includes the following figure supplement(s) for figure 1:

**Figure supplement 1.** CARD14 expression increases with differentiation in human and mouse keratinocytes.
**Figure supplement 2.** Conditional expression of CARD14$^{E138A}$ in *Card14*$^{LSL-E138A/+}$ *Rosa26*$^{CreERT2/+}$ mice induces rapid changes in back skin histology.
**Figure supplement 3.** Flow cytometry gating strategy used to identify different immune cell populations FACS plots show cells in the ears of one representative *Card14*$^{LSL-E138A/+}$ *Rosa26*$^{CreERT2/+}$ mouse at 1 m following the first tamoxifen injection.

## CARD14$^{E138A}$ signalling in keratinocytes induces skin inflammation independently of the adaptive immune system

CARD14 is expressed at high levels in skin keratinocytes but is also expressed at lower levels in intestinal epithelial cells and other tissues (**Figure 1A**). To test whether the skin inflammation in the *Card14*$^{LSL-E138A/+}$ *Rosa26*$^{CreERT2/+}$ mice was caused by keratinocyte-intrinsic CARD14$^{E138A}$ signalling, *Card14*$^{LSL-E138A/LSL-E138A}$ mice were crossed with *Krt14*$^{CreERT2}$ mice. Tamoxifen injection of the resulting *Card14*$^{E138A/+}$ *Krt14*$^{CreERT2/+}$ mice induced expression of CARD14$^{E138A}$ in keratinocytes (**Figure 2—figure supplement 1A**). This induced acanthosis, hyperkeratosis, infiltration of S100A9$^+$ myeloid cells and increased expression of chemokines and proinflammatory cytokines in the ears and the back skin at 5d after tamoxifen injection (**Figure 2A,B,C,D** and **Figure 2—figure supplement 1B,C,D and E**). Some *Card14*$^{E138A/+}$ *Krt14*$^{CreERT2/+}$ mice presented a visible skin phenotype without tamoxifen injection, which could be due to leaky expression of Cre allowing basal expression of CARD14$^{E138A}$, these were excluded from experiments. Skin expression of psoriasis-associated genes at 5d was increased in *Card14*$^{LSL-E138A/+}$ *Krt14*$^{CreERT2/+}$ mice compared to controls, similar to *Card14*$^{LSL-E138A/+}$ *Rosa26*$^{CreERT2/+}$ mice (**Figure 2—figure supplement 2A**). However, the fold changes for the majority of these transcripts were higher on the *Rosa26*$^{CreERT2}$ background. Nevertheless, these experiments showed that CARD14$^{E138A}$ signalling in keratinocytes alone was sufficient to induce skin inflammation.

To investigate the role of the adaptive immune system in the development of CARD14$^{E138A}$-induced skin pathology, *Card14*$^{LSL-E138A/+}$ *Rosa26*$^{CreERT2/+}$ *Rag1*$^{-/-}$ mice were generated that lacked T and B cells. Tamoxifen induced a skin phenotype very similar to *Card14*$^{LSL-E138A/+}$ *Rosa26*$^{CreERT2/+}$ *Rag1*$^{+/+}$ controls, with acanthosis, hyperkeratosis, S100A9$^+$ myeloid cell infiltration and expression of chemokines and proinflammatory cytokines at 5d following tamoxifen injection (**Figure 2E,F,G,H** and **Figure 2—figure supplement 1F,G,H and I**). Consequently, the adaptive immune system was not required for acute skin inflammation induced by ubiquitous CARD14$^{E138A}$ expression.

## Neutralisation of TNF efficiently blocks skin inflammation induced by CARD14$^{E138A}$

Two of the most common and efficient treatments for moderate to severe psoriatic patients are antibody blocking of TNF and IL-17A (**Greb et al., 2016**). We next assessed whether the inflammatory skin phenotype induced by CARD14$^{E138A}$ could be ameliorated by neutralising either of these cytokines following the protocol shown in **Figure 3A**.

Anti-TNF substantially reduced acanthosis in ears and back skin of tamoxifen-induced *Card14*$^{LSL-E138A/+}$ *Rosa26*$^{CreERT2/+}$ mice such that epidermal thickness was similar to that detected in *Card14*$^{+/+}$ *Rosa26*$^{CreERT2/+}$ control mice after tamoxifen injection (**Figure 3B** and **Figure 3—figure supplement 1A**). Skin expression of mRNAs encoding chemokines and proinflammatory cytokines was also significantly reduced by anti-TNF in the ears and the back skin (**Figure 3C** and **Figure 3—figure supplement 1B**). Flow cytometric analyses demonstrated that anti-TNF reduced infiltration of neutrophils, DCs and eosinophils into the ears (**Figure 3D**).

Expression of *Card14* mRNA in the skin of *Card14*$^{LSL-E138A/+}$ *Rosa26*$^{CreERT2/+}$ mice 5d after tamoxifen injection was preferentially localised to the suprabasal layers of the epidermis (**Figure 3—figure**

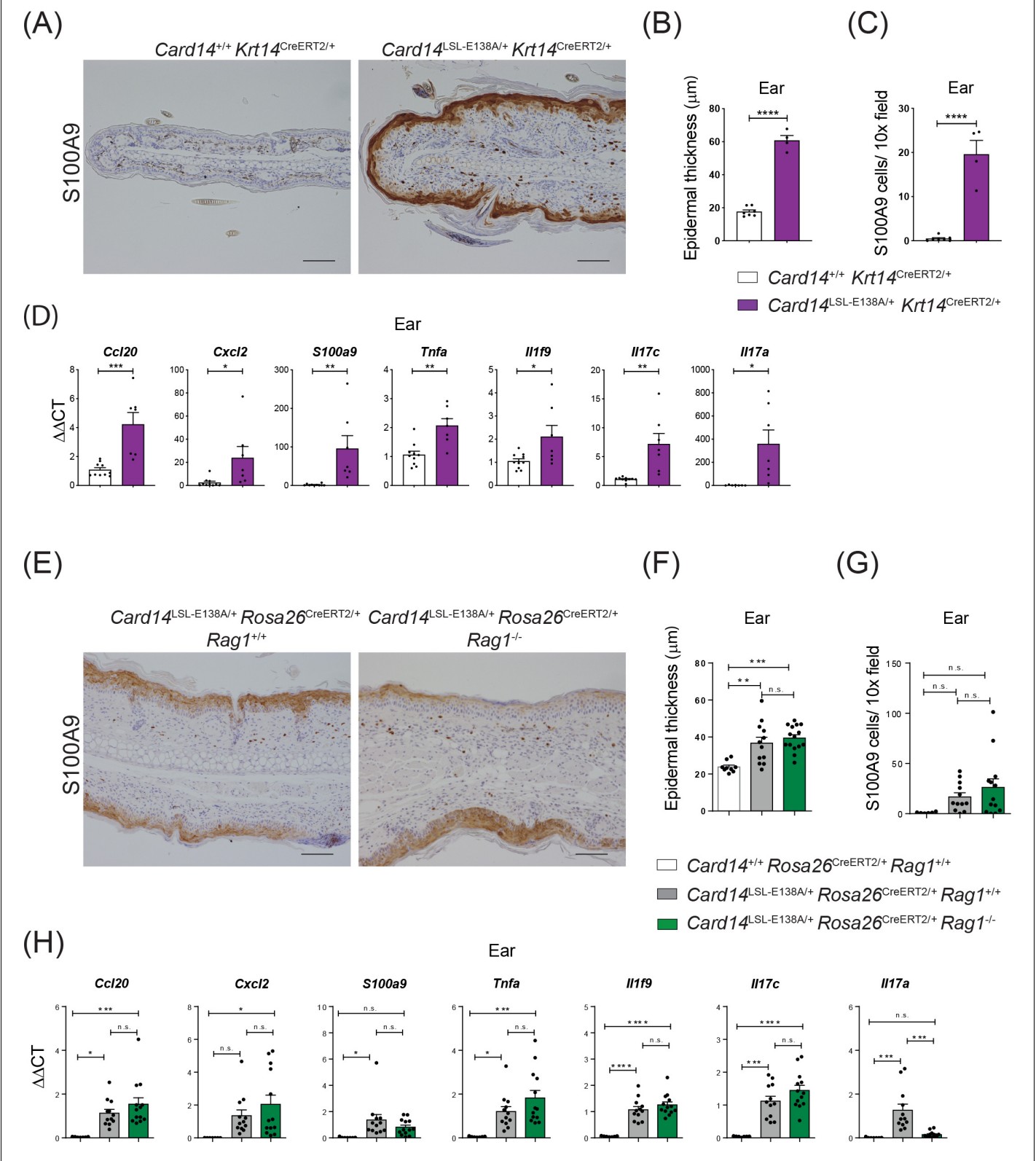

**Figure 2.** CARD14[E138A] signalling in keratinocytes induces ear skin inflammation independently of the adaptive immune system. Eight-week old Card14[+/+] Krt14[CreERT2/+], Card14[LSL-E138A/+] Krt14[CreERT2/+], Card14[LSL-E138A/+] Rosa26[CreERT2/+] Rag1[+/+], Card14[+/+] Rosa26[CreERT2/+] Rag1[+/+] and Card14[LSL-E138A/+] Rosa26[CreERT2/+] Rag1[-/-] mice were injected intraperitoneally with tamoxifen on days 0, 1 and 2. Mice were sacrificed and tissues analysed at 5d after the first injection of tamoxifen. (**A and E**) Immunohistochemistry of the ear stained for S100A9 at d5. (**B and F**) Quantification of acanthosis (**C and**

*Figure 2 continued on next page*

Figure 2 continued

G) Quantification of dermal immune cell infiltration. (**D and H**) qRT-PCR analysis of mRNA expression of the indicated genes in the ear. Fold changes in (**H**) were calculated by comparison with the *Card14*<sup>LSL-E138A/+</sup> *Rosa26*<sup>CreERT2/+</sup> *Rag1*<sup>+/+</sup> group. (**A and E**) Scale bar = 100 µm. Data in B and C from 1 of 2 similar experiments with n ≥ 4, data in D, F, G and H pooled from 2 independent experiments with n ≥ 4 (**D**) and with n ≥ 6 (**F, G and H**). Differences between groups analysed by Student's t test (**B, C and D**) or one-way ANOVA (**F, G and H**). *, $p<0.05$; **, $p<0.01$; ***, $p<0.001$; ****, $p<0.0001$. The online version of this article includes the following figure supplement(s) for figure 2:

**Figure supplement 1.** CARD14$^{E138A}$ expression in keratinocytes induces skin inflammation independently of the adaptive immune system.

**Figure supplement 2.** Ubiquitous and keratinocyte-specific expression of CARD14$^{E138A}$ promotes similar changes in gene expression in the skin.

---

*supplement 2A*). RNAscope analysis showed that *Tnfa* mRNA had a staining pattern resembling that of *Card14* mRNA (*Figure 3—figure supplement 2B*). Co-staining with specific antibodies revealed that *Tnfa* mRNA was largely produced by keratinocytes, with some contribution by neutrophils (*Figure 3—figure supplement 3A*). *Tnfa* mRNA was not detected in T cells in the skin. These results suggest that keratinocytes are the main source of TNF driving skin inflammation.

Blocking of IL-17A also resulted in a significant reduction in epidermal thickness of the back skin, but did not significantly reduce epidermal thickness in the ears (*Figure 3—figure supplement 1A* and *Figure 3B*). The expression of pro-inflammatory cytokines and chemokines in the ears was not altered by anti-IL-17A, although *S100a9* mRNA was reduced (*Figure 3C*). Anti-IL-17A reduced *Cxcl3* and *S100a9* mRNAs in the back skin of *Card14*<sup>LSL-E138A/+</sup> *Rosa26*<sup>CreERT2/+</sup> mice, but did not significantly alter the mRNA expression of the other cytokines and chemokines assayed (*Figure 3—figure supplement 1B*).

Together these results indicated that the acute inflammatory skin phenotype induced by tamoxifen injection of *Card14*<sup>LSL-E138A/+</sup> *Rosa26*<sup>CreERT2/+</sup> mice was predominantly driven by TNF produced by keratinocytes, while IL-17A appeared to play a less important role. These findings were consistent with the lack of requirement for adaptive immune cells for *Card14*$^{E138A}$ to induce acanthosis and expression of pro-inflammatory genes in the skin, since αβ and γδ T cells are a major source of IL-17A in psoriasis (*Prinz et al., 2020*). Consistent with this, *Il17a* mRNA levels were not increased in ears of *Card14*<sup>LSL-E138A/+</sup>*Rosa26*<sup>CreERT2/+</sup>*Rag1*<sup>-/-</sup> mice following tamoxifen induction (*Figure 2H*).

## Comparison of the inflammatory effects of acute versus chronic CARD14$^{E138A}$ expression

Histological and flow cytometric analyses of ears revealed clear evidence of acanthosis and skin inflammation, respectively, 5d following tamoxifen injection of *Card14*<sup>LSL-E138A/+</sup> *Rosa26*<sup>CreERT2/+</sup> mice (*Figure 1*), but externally the ears appeared normal (*Figure 4A*). However, one month (1 m) after CARD14$^{E138A}$ induction, large scales with sharply demarcated edges were clearly evident on the ears of the mutant mice (*Figure 4A*). No obvious external back skin phenotype was visible at 5d or 1 m. Histologic examination of ears revealed increased acanthosis at 1 m compared with 5d (*Figure 4B*). In contrast, epidermal thickening of the back skin, although still detected, was significantly reduced at 1 m compared to 5d (*Figure 4B*).

Flow cytometry was used to compare immune cell infiltration in the ears and back skin at 5d and 1 m post-tamoxifen injection (*Figure 4C*). At 5d, the immune cell infiltrates in the ears and skin was largely composed of innate cells, with substantially increased numbers of neutrophils, macrophages, dendritic cells and NK cells. The number of αβ T cells was only modestly increased (approximately two-fold) at 5d in ears, but not altered in back skin. However, the numbers of αβ T cells, including Tregs, were substantially increased in both ears and back skin at 1 m. CARD14$^{E138A}$ expression did not increase numbers of γδ T cells in ears or back skin at either time point.

Analysis of serum levels of TNF, IL-6 and IL-17A also revealed clear differences between 5d and 1 m post tamoxifen injection of *Card14*<sup>LSL-E138A/+</sup> *Rosa26*<sup>CreERT2/+</sup> mice. Each of these cytokines was elevated at 5d, but reduced to WT levels at 1 m (*Figure 4D*). Paralleling the early systemic overproduction of pro-inflammatory cytokines, *Card14*<sup>LSL-E138A/+</sup> *Rosa26*<sup>CreERT2/+</sup> mice suffered a decrease in body weight between 6-7d post-tamoxifen, returning to control levels by 10d (*Figure 4E*).

To investigate the role of the adaptive immune system in skin pathology induced by prolonged CARD14$^{E138A}$ signalling, *Card14*<sup>LSL-E138A/+</sup>*Rosa26*<sup>CreERT2/+</sup>*Rag1*<sup>-/-</sup> mice were analysed 1 m after tamoxifen injection. The absence of an adaptive immune system did not decrease ear and skin

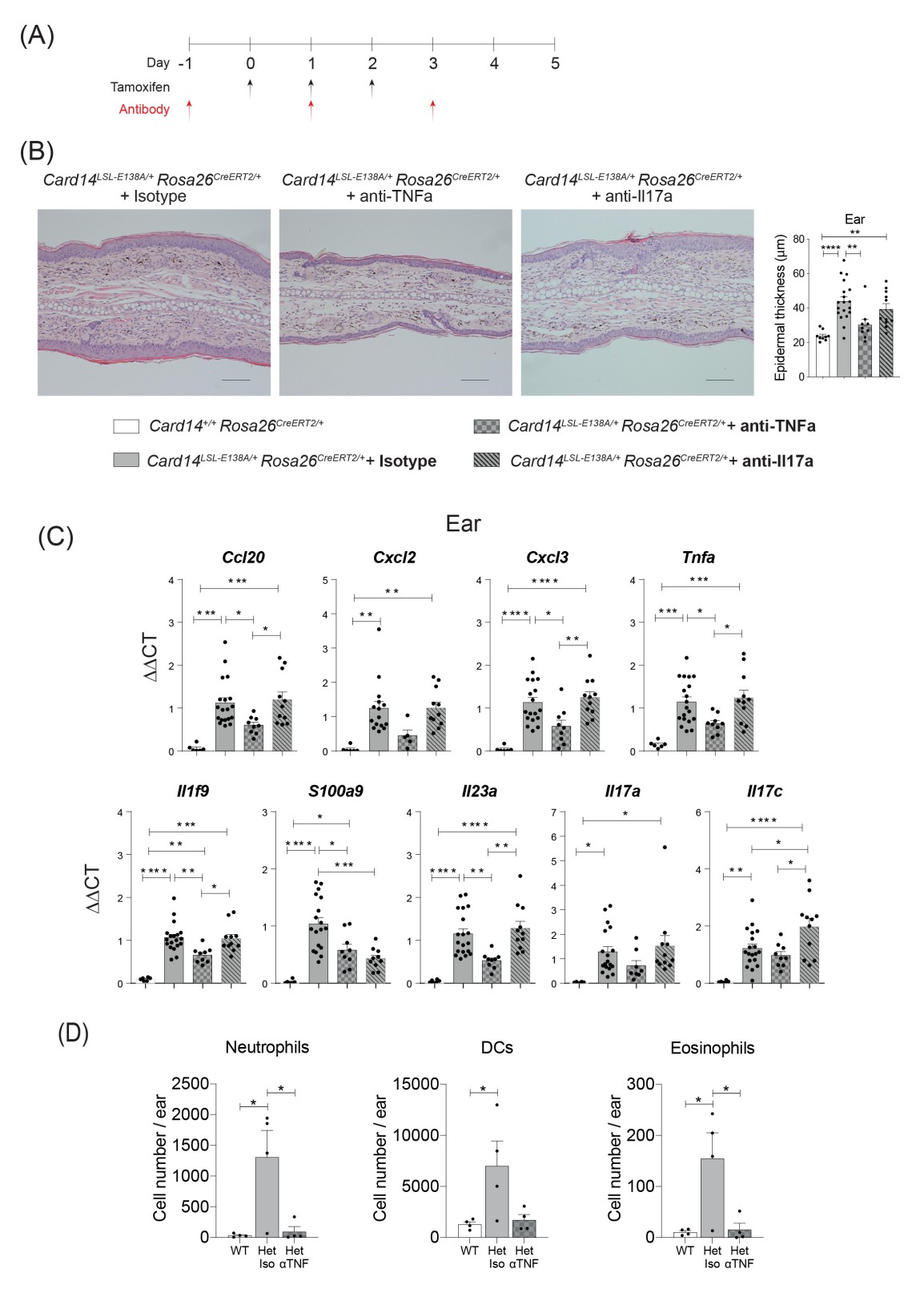

**Figure 3.** CARD14[E138A]-induced ear skin inflammation is ameliorated by anti-TNF. Eight-week old *Card14*[LSL-E138A/+]*Rosa26*[CreRT2/+] and *Card14*[+/+] *Rosa26*[CreRT2/+] mice treated with either tamoxifen and blocking antibodies or with tamoxifen and isotype control IgG following the scheme in **A**. (**B**) Representative histological image of ear stained by H and E on d5 and acanthosis quantification from mice receiving the different treatments. (**C**) qRT-PCR analysis of the expression of indicated genes Fold changes were calculated by comparing with the *Card14*[LSL-E138A/+] *Rosa26*[CreRT2/+] isotype

*Figure 3 continued on next page*

*Figure 3 continued*

treated group. (D) Quantification and characterisation of the immune cell infiltrate of the ears at d5 after tamoxifen by FACS. Data from 1 experiment (B) Scale bar = 100 μm. Data pooled from 2 experiments with at least n ≥ 4. Differences between groups analysed by one-way ANOVA. *, p<0.05; **, p<0.01; ***, p<0.001; ****, p<0.0001. For clarity, only statistically significant differences have been noted in the graphs.

The online version of this article includes the following figure supplement(s) for figure 3:

**Figure supplement 1.** CARD14$^{E138A}$-induced back skin inflammation is ameliorated by anti-TNF.
**Figure supplement 2.** Expression of *Tnfa* mRNA in the epidermis of *Card14*$^{LSL-E138A/+}$ *Rosa26*$^{CreERT2/+}$ at day 5 following tamoxifen induction.
**Figure supplement 3.** Keratinocytes are the main source of *Tnfa* mRNA in the epidermis of *Card14*$^{LSL-E138A/+}$ *Rosa26*$^{CreERT2/+}$ at day 5.

acanthosis, the abundance of S100A9$^{+}$ myeloid cells or the levels of proinflammatory mRNA transcripts induced by CARD14$^{E138A}$ (*Figure 4—figure supplement 1A–H*). Indeed, despite the absence of T and B cells, *Card14*$^{LSL-E138A/+}$*Rosa26*$^{CreERT2/+}$*Rag1*$^{-/-}$ mice compared to *Card14*$^{LSL-E138A/+}$*Rosa26*$^{CreERT2/+}$*Rag1*$^{+/+}$ controls had significantly higher numbers of infiltrating NK, eosinophils and monocytes, while numbers of neutrophils and macrophages were similar (*Figure 4—figure supplement 1I*). Thus, the adaptive immune system was not required for skin inflammation induced by chronic CARD14$^{E138A}$ signalling.

Taken together, these results suggest that acute CARD14$^{E138A}$ signalling in keratinocytes induced the transient systemic release of pro-inflammatory mediators that produced body weight loss. This early inflammatory burst was accompanied by a skin infiltrate predominantly composed of innate immune cells. After this acute phase, the skin lesion evolved and external ear skin morphology became similar to plaque psoriasis in humans. Chronic CARD14$^{E138A}$ signalling involved a prominent influx of T cells, although this was not required for induction of skin inflammation (from analysis of *Card14*$^{LSL-E138A/+}$*Rosa26*$^{CreERT2/+}$*Rag1*$^{-/-}$ mice), and reduced levels of pro-inflammatory cytokines systemically. The progression of the pathology in the *Card14*$^{LSL-E138A/+}$ *Rosa26*$^{CreERT2/+}$ mice resembled the flares suffered by psoriatic patients, which eventually undergo spontaneous resolution (*Greb et al., 2016*).

## CARD14$^{E138A}$ signalling induces dynamic changes in the skin transcriptome

To gain more insight into the mechanisms by which CARD14$^{E138A}$ induced skin inflammation, RNA was extracted from the ears of *Card14*$^{LSL-E138A/+}$*Rosa26*$^{CreERT2/+}$ and *Card14*$^{+/+}$*Rosa26*$^{CreERT2/+}$ mice 5d and 1 m after tamoxifen injection and subjected to RNA sequencing analysis. A principal component analysis indicated that the main difference between sample groups was the genotype (PCA1); CARD14$^{E138A}$ expression induced a genetic response clearly differentiated from the control groups (*Figure 5A*). The gene responses in the mutant mice were clearly distinct between 5d and 1 m, resulting in signalling duration being identified as the second cause of variation (PCA2) between sample groups (*Figure 5A*).

458 genes were found to be upregulated and 88 genes downregulated more than 2-fold in the *Card14*$^{LSL-E138A/+}$*Rosa26*$^{CreERT2/+}$ compared to the controls at 5d. Interestingly, the pattern of gene expression changed between 5d and 1 m, as shown by the different colour intensities in the heat map, indicating a change in transcriptional pattern with prolonged CARD14$^{E138A}$ signalling (*Figure 5B*). Pathway analysis revealed that TNF signalling via NF-κB was one of the pathways most strongly upregulated by CARD14$^{E138A}$ at both 5d and 1 m (*Figure 5C*). Strikingly, pathways related to extracellular matrix formation were also largely reduced in the *Card14*$^{LSL-E138A/+}$*Rosa26*$^{CreERT2/+}$ mice at 5d compared to the other groups. Expression of keratinisation genes was enriched in the *Card14*$^{LSL-E138A/+}$*Rosa26*$^{CreERT2/+}$ mice relative to controls, especially at 1 m.

Among the genes more strongly upregulated by CARD14$^{E138A}$ at 5d compared to 1 m were *Il6*, *Il1a*, *Osm*, *Mmp8*, *Il22*, *Ccl20*, Cxcl5, *Ccl17* and *Il17c* (*Figure 5D*). Several collagen genes (*Col11a1*, *Col6a6*, *Col13a1*, *Col14a1* and *Col16a1*) were downregulated by CARD14$^{E138A}$ at 5d and returned to WT levels by 1 m. These changes may be due to the effects of IL-17A (and possibly IL-17C) on the extracellular matrix (*Nakashima et al., 2012*). Cornification genes (*Klk8*, *Casp14*, *Flg* and *Krt6a*) were upregulated at 1 m compared to 5d, which was likely a secondary effect of chronic CARD14$^{E138A}$ signalling. *Krt6a* is implicated in psoriasis pathogenesis and is associated with the wound healing/regenerative phenotype seen in psoriasis (*Lessard et al., 2013*). Genes related to immune

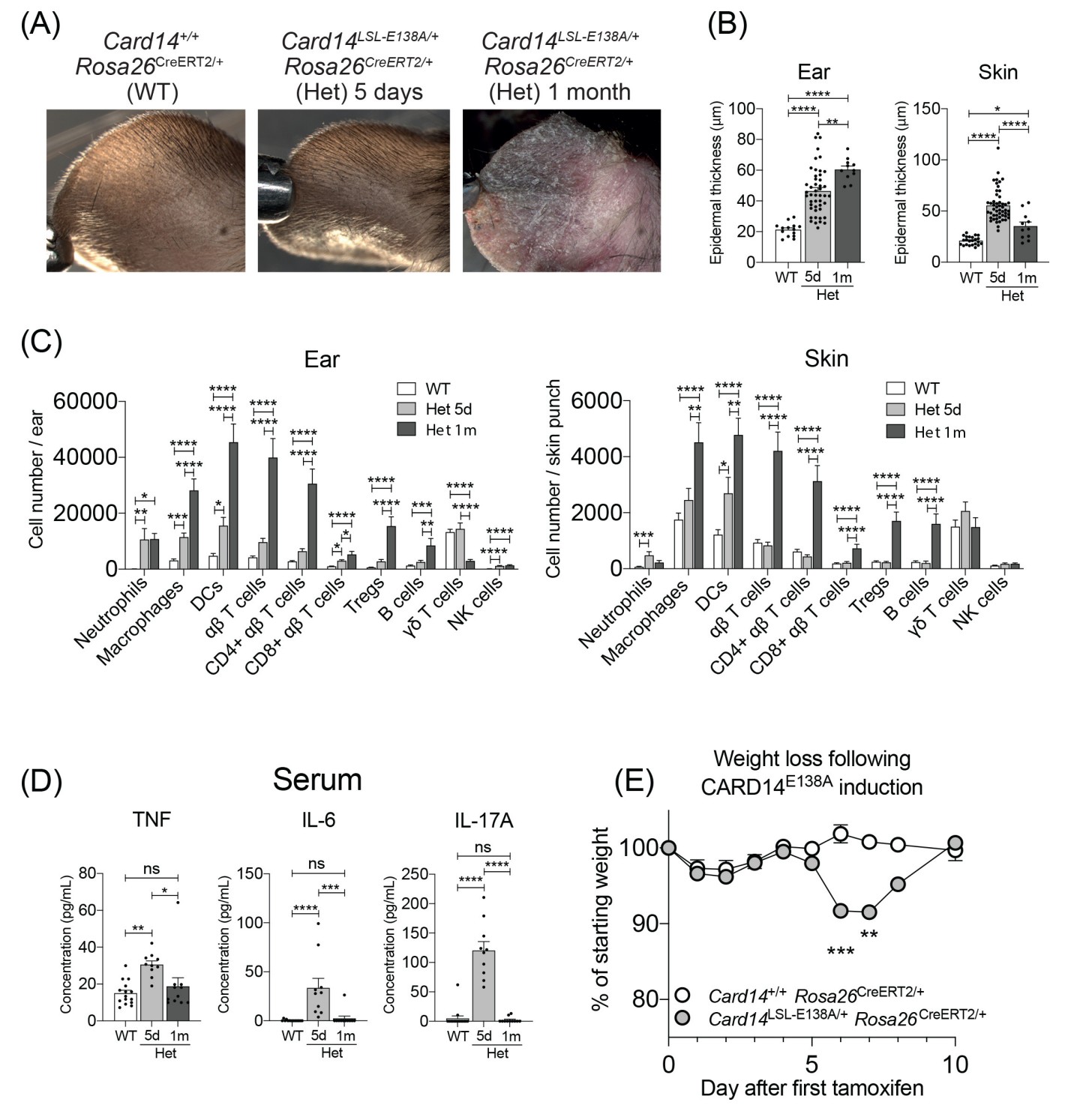

**Figure 4.** Evolution of inflammatory disease in *Card14*^LSL-E138A/+ *Rosa26*^CreERT2/+ mice over time. *Card14*^LSL-E138A/+*Rosa26*^CreERT2/+ mice (Het) and *Card14*^+/+ *Rosa26*^CreERT2/+ controls (WT) were intraperitoneally injected with tamoxifen on days 0, 1 and 2. Mice were sacrificed and tissues analysed at 5d or 1 m following the first tamoxifen injection. (**A**) The backs of the ears were photographed. Photos shown are representative of numerous mice. (**B**) Epidermal thickness of ear and skin was measured from H and E stained tissue sections. Data shown are a pool of 14 independent experiments. (**C**) Flow cytometry was performed on ear and skin. Ear: data pooled from 6 independent experiments: WT, n = 31; Het 5d, n = 22; Het 1 m, n = 13. Skin: data pooled from 9 independent experiments: WT, n = 43; Het 5d, n = 32; Het 1 m, n = 18. Note: FACS data for d5 are the same as that shown in *Figure 1F* (**D**) Serum was collected and cytokine concentrations analysed by immunoplex array. Serum samples collected from 3 independent

*Figure 4 continued on next page*

Figure 4 continued

experiments. (E) Weight was monitored over time. Card14$^{+/+}$ Rosa26$^{CreERT2/+}$ (n = 5), Card14 $^{LSL-E138A/+}$ Rosa26$^{CreERT2/+}$ (n = 6). For each day, differences between groups analysed by Student's t-test. Data collected from a mixture of male and female mice. (B-D) Differences between WT, Het 5d, and Het 1 m were analysed by one-way ANOVA. *, p<0.05; **, p<0.01; ***, p<0.001; ****, p<0.0001.

The online version of this article includes the following figure supplement(s) for figure 4:

**Figure supplement 1.** Lack of adaptive immune cells does not alter long-term pathology induced by heterozygous ubiquitous expression of CARD14$^{E138A}$.

cells changed in line with the types of immune cell infiltrates seen at different time points (*Figure 4C*). *Ly6g* (a marker for neutrophils) was markedly upregulated at 5d, while markers for T cells (*Cd4*, *Cd8a* and *Foxp3*) were only upregulated at 1 m.

Together the results in this section indicated that conditional expression of CARD14$^{E138A}$ in adult mice allowed the evolution of CARD14$^{E138A}$-induced skin inflammation to be monitored. This involved downregulation of some key proinflammatory genes at 1 m, which may reflect a homeostatic response to chronic activation of CARD14$^{E138A}$ signalling pathways. Importantly, several genes relevant to the pathology of psoriasis (*S100a9*, *Il1f9*, *Cxcl2*, *Il17a* and *Tnfa*) remained significantly elevated in *Card14*$^{LSL-E138A/+}$*Rosa26*$^{CreERT2/+}$ mice at 1 m and may be necessary in maintaining the inflammatory skin phenotype promoted by chronic CARD14$^{E138A}$ signalling.

## Comparison of the CARD14$^{E138A}$ skin transcriptome with human psoriatic skin transcriptome

To determine whether the transcriptional changes in the skin of *Card14*$^{LSL-E138/+}$ *Rosa26*$^{CreERT2/+}$ mice were similar to Psoriasis Vulgaris (PV), the CARD14$^{E138A}$ skin transcriptome was compared with the lesional skin transcriptome of psoriatic patients (*Tsoi et al., 2019*). This showed that the CARD14$^{E138A}$ transcriptome at 5d and 1 m (*Figure 5—figure supplement 1A*) had relatively low $R^2$ coefficients of determination with the PV datasets ($R^2$: 0.255 at 5d and $R^2$: 0.185 at 1 m). Nevertheless, when comparing the differentially expressed genes (DEG; logFC $\geq$2, p-adj $\leq$0.05) from the ears of *Card14*$^{LSL-E138/+}$ *Rosa26*$^{CreERT2/+}$ mice after tamoxifen injection and the skin of PV patients, 34% and 30% of genes at 5d and 1 m, respectively, were shared with PV (*Figure 5—figure supplement 1B*). This striking overlap suggested that the skin inflammation that develops in *Card14*$^{LSL-E138/+}$ *Rosa26*$^{CreERT2/+}$ mice after tamoxifen injection shares mechanistic similarities with PV.

Comparison of the PV skin transcriptome with the published skin transcriptome induced by constitutive ubiquitous expression of CARD14$^{\Delta E138}$ in mice (*Mellett et al., 2018*; *Figure 5—figure supplement 1C*) also showed only low correlation ($R^2$: 0.23), although the CARD14$^{\Delta E138}$-induced skin transcriptome was well correlated with that of *Card14*$^{LSL-E138/+}$ *Rosa26*$^{CreERT2/+}$ mice at both 5d ($R^2$: 0.566) and 1 m ($R^2$: 0.656) after tamoxifen injection (*Figure 5—figure supplement 1D*). This suggested that CARD14$^{E138A}$-induced skin inflammation in constitutive and inducible mouse models involved similar mechanisms.

The low correlation coefficients between transcriptomes of *Card14*$^{LSL-E138/+}$ *Rosa26*$^{CreERT2/+}$ mice and PV are likely caused by comparing between species (*Breschi et al., 2017*). Consistent with this, the tamoxifen-induced *Card14*$^{LSL-E138A/+}$ *Rosa26*$^{CreERT2}$ skin transcriptome at 5d was strongly correlated ($R^2$: 0.684) with the published IL-23-induced skin transcriptome in mice (*Gauld et al., 2018*; *Figure 5—figure supplement 1E*). The IL-23-induced psoriasiform dermatitis model is thought to most closely resemble human psoriasis (*Suárez-Fariñas et al., 2013*).

The *CARD14*$^{E138A}$ mutation was identified in a patient with GPP. Comparison of the transcriptomes from the skin of this patient with patients with classical psoriasis revealed similar differences when compared to controls, with higher transcript levels associated with the more severe form of psoriasis (*Jordan et al., 2012b*). Genes with logFC $\geq$2 (padj $\leq$0.05) in *Card14*$^{LSL-E138A/+}$ *Rosa26*$^{CreERT2/+}$ mice at 5d and 1 m after tamoxifen injection were compared with genes upregulated 2-fold or more in PV and GPP (*Johnston et al., 2017*). The skin transcriptome of *Card14*$^{LSL-E138A/+}$ *Rosa26*$^{CreERT2/+}$ mice shared more upregulated genes with GPP than PV at both 5d (7% and 3.5%, respectively) and 1 m (6.2% and 3.6%, respectively) (*Figure 5—figure supplement 2A*). Moreover, *Steap1* and *Steap4* mRNAs were found to be significantly increased in the ears of *Card14*$^{LSL-E138A/+}$ *Rosa26*$^{CreERT2/+}$ mice at 5d and 1 m following tamoxifen injection (*Figure 5—figure supplement 2B*). The

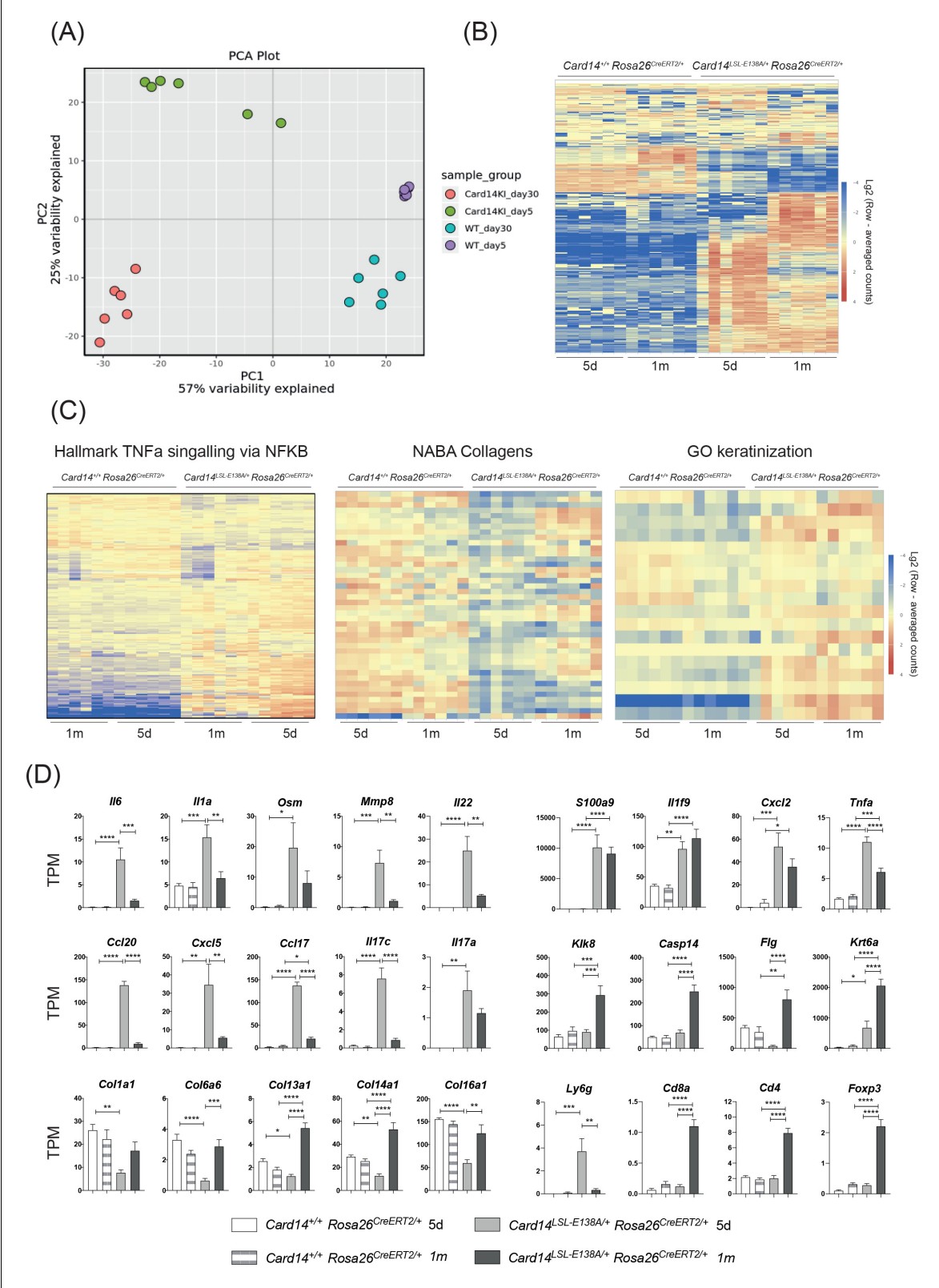

**Figure 5.** Acute and chronic effects of CARD14[E138A] signalling on gene transcription in the skin. *Card14*[LSL-E138A/+]*Rosa26*[CreERT2/+] mice (Het) and *Card14*[+/+] *Rosa26*[CreERT2/+] controls (WT) were intraperitoneally injected with tamoxifen on days 0, 1 and 2. Mice were sacrificed and tissues analysed at 5d or 1 m following the first tamoxifen injection (each experimental group n = 6). (**A**) Principal Component analysis separated samples by genotype (PCA1) and time point (PCA2). (**B**) Heatmap showing the top 500 differentially expressed genes. (**C**) Heatmap of the genes contained in the TNFα

*Figure 5 continued on next page*

*Figure 5 continued*

signalling via NF-κB (left), Naba Collagens (middle) and Keratinisation (right) pathways. (**D**) Graphs showing the expression levels of individual genes. TPM (Transcripts Per Million). Differences between WT, Het 5d and Het 1 m were analysed by one-way ANOVA. *, p<0.05; **, p<0.01; ***, p<0.001; ****, p<0.0001.

The online version of this article includes the following figure supplement(s) for figure 5:

**Figure supplement 1.** Similarities between the skin transcriptomes of *Card14*^LSL-E138A/+ *Rosa26*^CreERT2/+ mice and human skin pathologies.
**Figure supplement 2.** The skin transcriptome of *Card14*^LSL-E138A/+*Rosa26*^CreERT2/+ mice is more similar to the skin transcriptome of GPP than PV.
**Figure supplement 3.** The skin transcriptome of *Card14*^LSL-E138A/+*Rosa26*^CreERT2/+ mice is more similar to the skin transcriptome of PV than AD.

expression of *STEAP1* and *STEAP4* mRNAs is elevated in three different pustular skin disorders in humans and thought to be important in the regulation of proinflammatory neutrophil-activating cytokines (*Liang et al., 2017b*). The minor involvement of the adaptive immune system in skin inflammation induced by CARD14^E138A and these transcriptomic similarities suggested that *Card14*^LSL-E138A/+ *Rosa26*^CreERT2/+ mice might model some aspects of GPP.

Atopic dermatitis (AD) is the most common inflammatory skin disorder, driven by both terminal keratinocyte differentiation defects and strong type two immune responses (*Guttman-Yassky and Krueger, 2017*). It has been proposed that loss-of-function mutations in *CARD14* are associated with severe AD (*Peled et al., 2019*), in contrast to the gain-of-function *CARD14* mutations that promote psoriasis. Comparison of the CARD14^E138A-induced skin transcriptome with the lesional skin transcriptome of atopic dermatitis (AD) patients (*Tsoi et al., 2019*) revealed only a weak correlation at 5d (R$^2$: 0.221) and 1 m (R$^2$: 0.17), similar to comparisons with the PV transcriptome and probably mainly reflecting mouse/human differences (*Figure 5—figure supplement 3A*).

To gain a clearer insight into whether skin inflammation in *Card14*^LSL-E138/+ *Rosa26*^CreERT2/+ mice resembled AD, DEG (logFC ≥2, padj ≤0.05) in *Card14*^LSL-E138A/+ *Rosa26*^CreERT2/+ mice were compared with gene expression levels in normal, non-lesional and lesional skin of PV and AD patients (*Figure 5—figure supplement 3B*). Heat maps showed that a significant fraction of 5d and 1 m CARD14^E138A upregulated genes were highly expressed in PV patients (red genes). However, these genes were expressed at lower levels in AD patients. In addition, *IL17a*, *IL19*, *IL36a* (*Swindell et al., 2016*) and *IL36g* (*D'Erme et al., 2015*), which are considered to be biomarkers of psoriatic skin, were all strongly upregulated in the skin of *Card14*^LSL-E138A/+ *Rosa26*^CreERT2/+ (*Figure 5D* and *Figure 5—figure supplement 3C*). In contrast, AD involves downregulation of cornification proteins (eg. *FLG* and *IVL*) and upregulation of the TH2 cytokines *IL4, IL5, IL33, TSLP* and *CCL24* (*Guttman-Yassky and Krueger, 2017*), which are not changed or follow opposite trends in *Card14*^LSL-E138A/+ *Rosa26*^CreERT2/+ mice (*Figure 5D* and *Figure 5—figure supplement 3C*). These comparative analyses were consistent with CARD14^E138A inducing a psoriasis-like skin inflammation and not skin inflammation with gene expression characteristics of AD.

## Homozygous expression of *Card14*^E138A induces a severe systemic illness

The majority of psoriasis-associated *CARD14* mutations identified are heterozygous suggesting that homozygous expression of such CARD14 mutations might exceed a threshold of NF-κB activation that is compatible with life. To investigate whether the inflammatory effects of CARD14^E138A were dose dependent, the responses of *Card14*^LSL-E138A/LSL-E138A*Rosa26*^CreERT2/+ (homozygous) mice to tamoxifen injection were compared with *Card14*^LSL-E138A/+*Rosa26*^CreERT2/+ (heterozygous) mice.

Tamoxifen injection of *Card14*^LSL-E138A/+*Rosa26*^CreERT2/+ mice did not result in any significant weight changes up to 4d (*Figure 6A*). In contrast, *Card14*^LSL-E138A/LSL-E138A *Rosa26*^CreERT2/+ mice started losing weight around 3d after tamoxifen induction. By 5d, homozygous mice had lost near to 20% of their initial body weight (*Figure 6A*), reaching humane end-points and requiring sacrifice. Upon general inspection, homozygous mice at d5 appeared scaly, emaciated, inactive and hunched, while heterozygous mice remained in good health.

The fraction of body surface covered by visible scales appeared to be greater in the homozygous mice than their heterozygous littermates. The middle and lower back skin of the heterozygous mice was largely clear of visible scaling, although areas of skin that were easily accessible to the mice (upper back and beneath the chin) often presented fine scales. In contrast, large areas of back skin

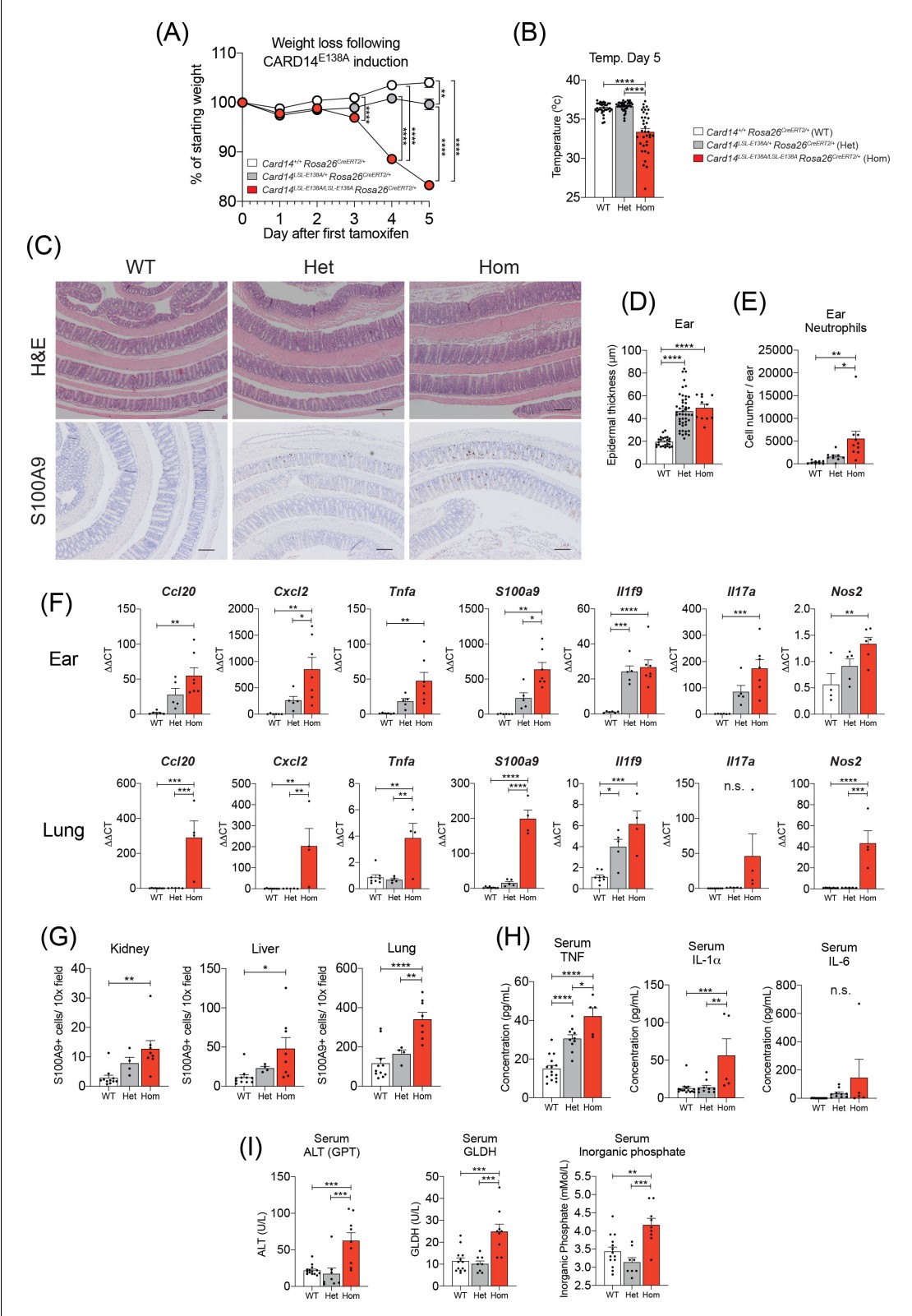

**Figure 6.** Induction of CARD14^E138A in *Card14*^LSL-E138A/LSL-E138A *Rosa26*^CreERT2/+ mice induces a severe inflammatory phenotype. *Card14*^LSL-E138A/+ *Rosa26*^CreERT2/+ mice (Het), *Card14*^LSL-E138A/LSL-E138A *Rosa26*^CreERT2/+ mice (Hom) and *Card14*^+/+ *Rosa26*^CreERT2/+ controls (WT) were intraperitoneally injected with tamoxifen on d0, 1 and 2. All mice were sacrificed 5d after the first injection of tamoxifen and tissues collected for analysis. (**A**) Weight was monitored every day. WT (n = 24, from five independent experiments), Het (n = 22, from five independent experiments), Hom

*Figure 6 continued on next page*

*Figure 6 continued*

(n = 49, from nine independent experiments). (**B**) The temperatures of all mice were taken using a rectal thermometer on d5. WT from four independent experiments, Het from two independent experiments, Hom from six independent experiments. (**C**) Representative histology images of colons stained with H and E (upper panels) or anti-S100A9 (lower panels) are shown. (**D**) Epidermal thickness of ear at d5 was measured from H and E stained tissue sections. Data collected from 15 independent experiments. (**E**) Flow cytometry was performed to calculate neutrophil numbers in ears at d5. Data pooled from two independent experiments. (**F**) qRT-PCRs of d5 ear and lung tissue were performed. Fold changes were calculated by comparison with the WT mice group. Data pooled from two independent experiments. (**G**) S100A9 staining was performed via immunohistochemistry to enumerate myeloid cell numbers in kidney, liver, and lung at d5. Data pooled from two independent experiments. (**H**) Serum was taken at the point of sacrifice and analysed by immunoplex array. (**I**) Serum quantification of biochemistry for indicators of liver (ALT: alanine aminotransferase and GLDH: glutamate dehydrogenase) and kidney (inorganic phosphate) damage. Sera were collected from 4 (H and I) independent experiments. Data collected from a mixture of male and female mice. (B, D-I) Differences between WT, Het, and Hom analysed by one-way ANOVA. *, $p<0.05$; **, $p<0.01$; ***, $p<0.001$; ****, $p<0.0001$. (**C**) Scale bar = 200 µm.

The online version of this article includes the following figure supplement(s) for figure 6:

**Figure supplement 1.** Increased skin scaling in *Card14*$^{LSL-E138A/LSL-E138A}$ *Rosa26*$^{CreERT2/+}$ compared to *Card14*$^{LSL-E138A/+}$ *Rosa26*$^{CreERT2/+}$ mice following tamoxifen injection.

**Figure supplement 2.** Effects of CARD14$^{E138A}$ expression in the colon.

in the homozygous mice, including sections less easy to scratch, were covered in fine scales (*Figure 6—figure supplement 1A*). The skin of some homozygotes also appeared erythrodermic. The ears of the homozygous mice were visibly scalier than those of the heterozygotes, although the foot pads of both genotypes were similarly affected (*Figure 6—figure supplement 1B and C*). Of particular note, the lips of the d5 homozygous mice were consistently chapped (*Figure 6—figure supplement 1D*), while the lips of the heterozygous mice appeared normal.

On d5, homozygous mice were cold to the touch and measurement of rectal temperature confirmed hypothermia (*Figure 6B*). In contrast, rectal temperatures of heterozygous mice were similar to WT controls. Frequently, homozygous mice developed diarrhoea and dissection revealed that small and large intestines and caecum were filled with runny yellow fluid and occasional pockets of gas (*Figure 6—figure supplement 2A*). Despite this apparent gut phenotype, histological analysis of small intestine (not shown) and colon (*Figure 6C*) did not reveal any gross tissue damage. Although homozygous mice appeared emaciated and dehydrated, the stomachs were full of food and much enlarged. The livers of the homozygous mice often appeared unusually pale. Histological examination of the ear and lower back skin revealed a similar extent of acanthosis present in heterozygous and homozygous mice (*Figure 6D*). Hence, while body surface involvement, as measured by visible external scaling, was greater in the homozygous mice, epidermal thickening was comparable between genotypes.

Flow cytometry showed increased neutrophil infiltration into the ears of the homozygous mice as compared to the heterozygotes (*Figure 6E*). Moreover, qRT-PCR analysis of d5 ear mRNA revealed a gene dosage effect of *Card14*$^{E138A}$ for the majority of genes analysed and the expression of key psoriasis associated genes was higher in homozygous mice than in heterozygous mice (*Figure 6F*). Immunohistochemical analyses revealed significantly higher infiltration of S100A9$^+$ myeloid cells in kidney, liver, colon, and lung in homozygous mice compared to heterozygotes (*Figure 6C*, lower panels and 6G). The mRNA expression of several pro-inflammatory genes, including *Ccl20*, *Cxcl2*, *S100A9* and *Il17c*, was also significantly increased in the lungs of homozygous mice, but not heterozygous mice, compared to WT controls (*Figure 6F*). Analyses of sera demonstrated that circulating levels of TNF and IL-1α were increased in homozygous mice compared to heterozygotes and there was a trend toward increased IL-6 levels (*Figure 6H*). Similarly, serum ALT (alanine aminotransferase), GLDH (glutamate dehydrogenase) and inorganic phosphate were significantly elevated in the homozygous, but not heterozygous mice (*Figure 6I*). Increased circulating levels of ALT and GLDH are indicative of liver damage, while increased inorganic phosphate is indicative of defective kidney function (*Vervloet et al., 2017*). These results suggested multi-organ dysfunction in d5 homozygous mice.

In conclusion, the results in this section indicated that homozygous expression of *Card14*$^{E138A}$ in adult mice caused a severe systemic illness, in addition to widespread skin inflammation. This phenotype had striking similarities to the systemic phenotype displayed in exacerbations of GPP (*Bachelez, 2020*).

## Keratinocyte-intrinsic CARD14$^{E138A}$ signalling induces systemic inflammation independently of adaptive immune cells

Immunoblotting analyses demonstrated that the skin and gastrointestinal tract were the main sites of WT CARD14 expression (*Figure 1A*). RNAscope analysis of the colon revealed *Card14* expression to be largely restricted to the single layer of epithelial cells lining the gut lumen (*Figure 6—figure supplement 2B*). Furthermore, *Card14* mRNA expression was found to increase from proximal to distal ends of the colon (*Figure 6—figure supplement 2B–D*).

The colons of *Card14*$^{LSL-E138A/+}$ *Rosa26*$^{CreERT2/+}$ mice exhibited infiltration of S100A9$^+$ myeloid cells at 5d after tamoxifen and increased expression of mRNAs encoding proinflammatory cytokines (*Figure 6—figure supplement 2E–F*). Given the unusual gut phenotype observed in homozygous *Card14*$^{LSL-E138A/LSL-E138A}$*Rosa26*$^{CreERT2/+}$ mice following tamoxifen injection, it was possible that gut-specific expression of CARD14$^{E138A}$ was driving the severe systemic phenotype.

To investigate this, *Card14*$^{LSL-E138A/LSL-E138A}$ *Villin*$^{CreERT2}$ were generated to restrict CARD14$^{E138A}$ expression to the epithelial cells of the gut. Tamoxifen injection of the resulting *Card14*$^{LSL-E138A/LSL-E138A}$ *Villin*$^{CreERT2}$ mice, induced expression of CARD14$^{E138A}$ only in intestinal epithelial cells (*Figure 6—figure supplement 2G*). *Card14*$^{LSL-E138A/LSL-E138A}$ *Villin*$^{CreERT2}$ mice exhibited no gross phenotype at d5 after tamoxifen and were healthy (*Figure 7A*), with no signs of skin disease or organ dysfunction, and presented a histologically normal colon even 1 m after induction (*Figure 6—figure supplement 2H*). Tamoxifen injection induced expression of *Ccl20*, *Tnfa* and *Nos2* mRNAs in the gut of *Card14*$^{LSL-E138A/LSL-E138A}$ *Villin*$^{CreERT2}$ mice resulting from CARD14$^{E138A}$ signalling in gut epithelial cells (*Figure 6—figure supplement 2I*). These results suggested that CARD14$^{E138A}$ signalling in the gut was not sufficient to promote skin inflammation or the systemic inflammatory phenotype.

To determine whether CARD14$^{E138A}$ signalling in keratinocytes was driving the severe systemic phenotype, homozygous *Card14*$^{LSL-E138A/LSL-E138A}$*Krt14*$^{CreERT2}$ mice were generated. Tamoxifen injection of these mice rapidly induced a pronounced loss of weight, comparable to that in *Card14*$^{LSL-E138A/LSL-E138A}$*Rosa26*$^{CreERT2/+}$ mice (*Figure 7A*). *Card14*$^{LSL-E138A/LSL-E138A}$*Krt14*$^{CreERT2}$ mice also became hypothermic following tamoxifen administration and displayed significant infiltration of S100A9$^+$ myeloid cells in the lungs (*Figure 7B and C*). Upon dissection, these mice also showed signs of organ dysfunction, most notably swollen stomach and intestines (data not shown), akin to those seen in the *Card14*$^{LSL-E138A/LSL-E138A}$*Rosa26*$^{CreERT2/+}$ mice (*Figure 6—figure supplement 2A*). These results indicated that keratinocyte-intrinsic CARD14$^{E138A}$ signalling was sufficient to induce severe systemic inflammation.

To determine whether the adaptive immune system contributed to the development of systemic disease induced by homozygous CARD14$^{E138A}$ expression, *Card14*$^{LSL-E138A/LSL-E138A}$*Rosa26*$^{CreERT2/+}$ *Rag1*$^{-/-}$ mice were generated. The absence of T and B cells in these mice did not ameliorate the systemic phenotype induced by homozygous *Card14*$^{LSL-E138A}$, as judged by weight loss, hypothermia and extensive infiltration of S100A9$^+$ myeloid cells into the lungs (*Figure 7A–C*). Hence, the rapid development of systemic disease in homozygous *Card14*$^{LSL-E138A/LSL-E138A}$*Rosa26*$^{CreERT2/+}$ mice following tamoxifen injection was not dependent on the adaptive immune system.

## Anti-TNF ameliorates severe disease in homozygous *Card14*$^{LSL-E138A/LSL-E138A}$*Rosa26*$^{CreERT2/+}$ mice

Blocking TNF significantly reduced skin inflammation in heterozygous *Card14*$^{LSL-E138A/+}$*Rosa26*$^{CreERT2}$ mice following tamoxifen injection (*Figure 3*). *Card14*$^{LSL-E138A/LSL-E138A}$ *Rosa26*$^{CreERT2/+}$ mice were treated with anti-TNF blocking antibody prior to and post tamoxifen induction to determine whether the severe systemic disease was also mediated by TNF. Weight loss was significantly reduced by anti-TNF treatment compared to isotype control, but not completely blocked (*Figure 8A*). However, anti-TNF prevented the hypothermic response of *Card14*$^{LSL-E138A/LSL-E138A}$ *Rosa26*$^{CreERT2/+}$ mice to tamoxifen-induction (*Figure 8B*), which correlated with the mice appearing more active than isotype controls. Recruitment of S100A9$^+$ myeloid cells into the kidneys, liver and lung at d5 post-tamoxifen was also significantly reduced by anti-TNF compared to isotype control (*Figure 8C*).

Anti-TNF treatment significantly decreased ear mRNA expression of *Ccl20*, *Cxcl2* and *Cxcl3* (*Figure 8D*), the products of which are chemoattractants for neutrophils (*Chiang et al., 2019*). The reduction in expression of these genes by anti-TNF was even more pronounced in the lungs. mRNA

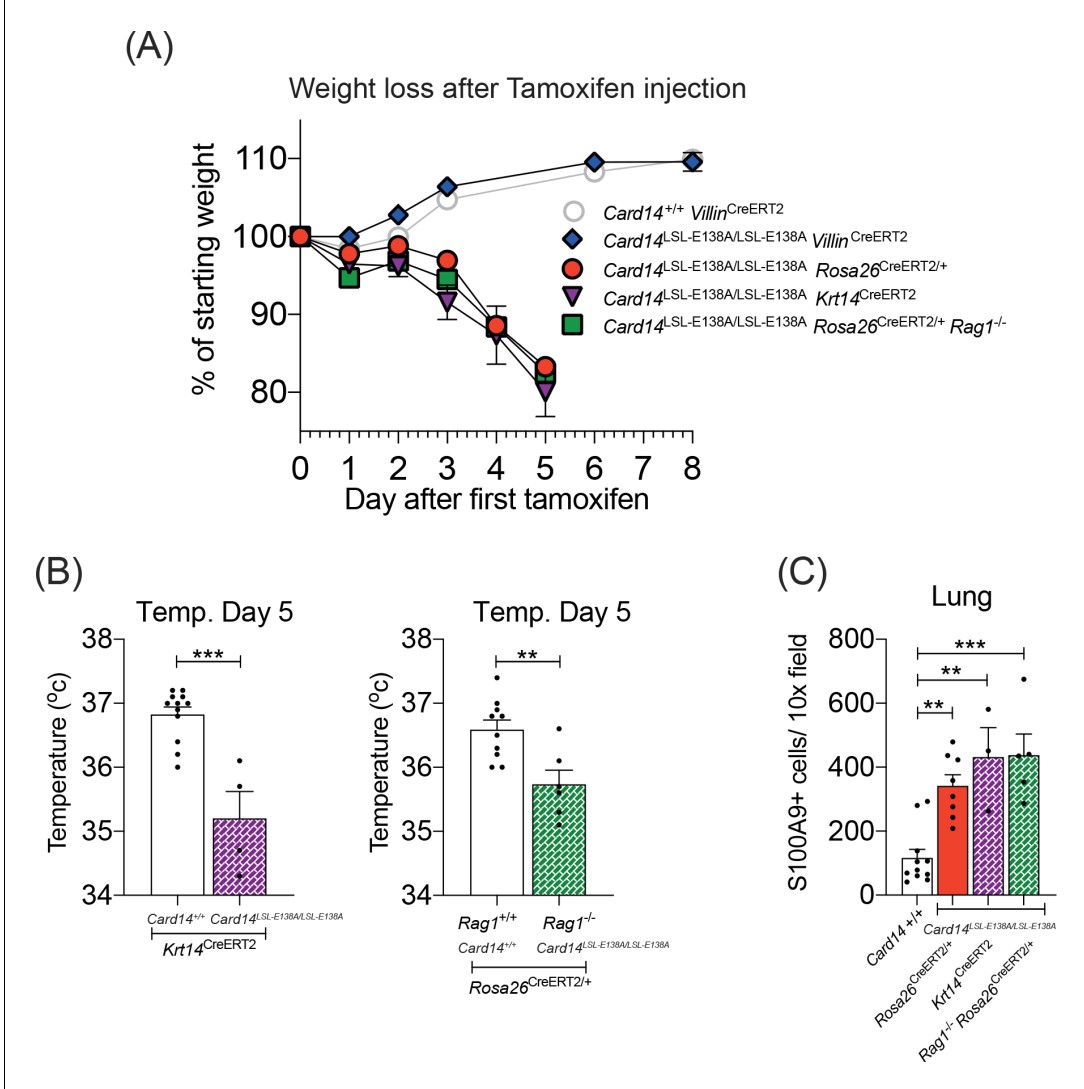

**Figure 7.** CARD14[E138A] signalling in keratinocytes induces a severe systemic phenotype independently of the adaptive immune system. Mice of indicated genotypes were intraperitoneally injected with tamoxifen on d0, 1 and 2. (**A**) Weight was monitored over time. Card14[LSL-E138A/LSL-E138A] Rosa26[CreERT2/+] (n = 49, from nine independent experiments), Card14[LSL-E138A/LSL-E138A] Krt14[CreERT2] (n = 6, from two independent experiments), and Rag1[-/-] Card14[LSL-E138A/LSL-E138A] Rosa26[CreERT2/+] mice (n = 11, from two independent experiments) lost weight, necessitating sacrifice 5d after the first injection of tamoxifen. Card14[LSL-E138A/LSL-E138A] Villin[CreERT2] mice (n = 7, from two independent experiments) and Card14[+/+] Villin[CreERT2] controls (n = 19, from four independent experiments) were monitored for an extended period and did not lose weight. (**B**) Temperatures of indicated mice were taken by rectal thermometer 5d after the first injection of tamoxifen. Card14[+/+] Krt14[CreERT2], and Card14[LSL-E138A/LSL-E138A] Krt14[CreERT2], from two independent experiments; Rag[+/+] Card14[+/+] Rosa26[CreERT2], from four independent experiments; Rag[-/-] Card14[LSL-E138A/LSL-E138A] Rosa26[CreERT2], from one experiment. Differences between groups analysed by Student's t-test. (**C**) Numbers of S100A9[+] myeloid cells were calculated from stained sections of lung tissue taken from indicated genotypes on d5. Card14[+/+] mice represent a mixture of Card14[+/+] Rosa26[CreERT2/+] and Card14[+/+] Krt14[CreERT2] mice. Data pooled from two independent experiments. Differences between groups analysed by one-way ANOVA. *, p<0.05; **, p<0.01; ***, p<0.001.

expression of *S100a9*, a marker of myeloid cell infiltration, was also significantly reduced (*Figure 8D*).

Reactive oxygen species (ROS) generation, degranulation and formation of NETs (neutrophil extracellular traps) by neutrophils in the skin contribute to the inflammatory pathogenesis of psoriasis (*Chiang et al., 2019*). Anti-TNF reduction in S100A9[+] myeloid cell infiltration into the liver, kidneys and lungs of Card14[LSL-E138A/LSL-E138A] Rosa26[CreERT2/+] mice suggested that reduction in myeloid cell-mediated inflammation might explain in part the protective effect of anti-TNF treatment. To investigate this, Card14[LSL-E138A/LSL-E138A] Rosa26[CreERT2/+] mice were administered anti-GR1 antibody, which depleted neutrophils and monocytes. Treatment with anti-GR1 before and during

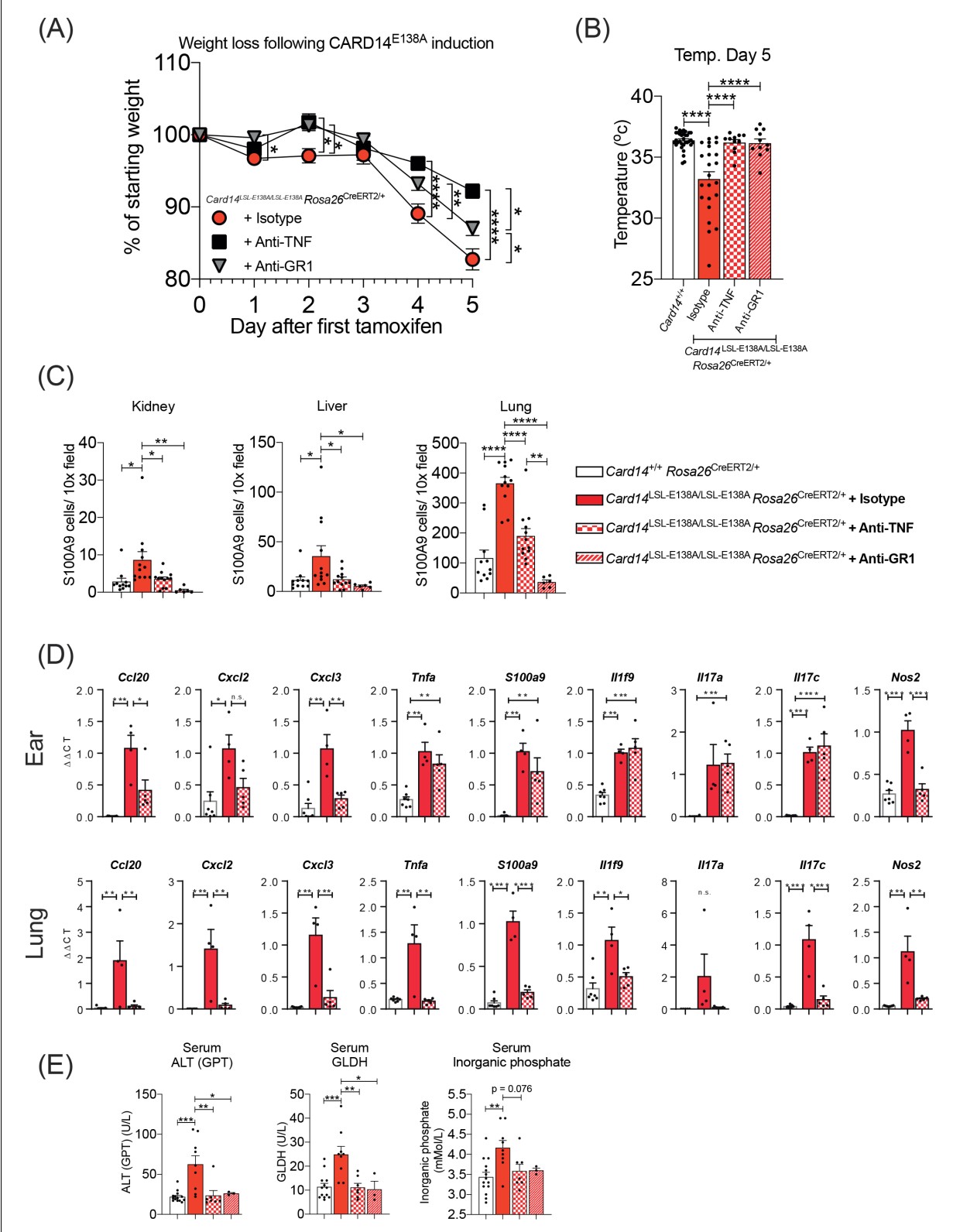

**Figure 8.** Anti-TNF ameliorates severe systemic phenotype induced in *Card14*^LSL-E138A/LSL-E138A^*Rosa26*^CreERT2/+^ mice *Card14*^LSL-E138A/LSL-E138A^*Rosa26*^CreERT2/+^ mice were intraperitoneally injected with tamoxifen on d0, 1 and 2 and intraperitoneally injected with either isotype control, anti-TNF, or anti-GR1 on days −1, 1 and 3. Mice were sacrificed on d5. (**A**) Mouse weight was monitored from d0 until the experiment end. Isotype (n = 24, from four independent experiments), anti-TNF (n = 11, from two independent experiments), anti-GR1 (n = 11, from two independent experiments). For

*Figure 8 continued on next page*

Figure 8 continued

each day, differences between groups were analysed by one-way ANOVA. (B) Mouse temperatures were taken by rectal thermometer on d5. $Card14^{+/+}$ control mice are a mixture of Cre-, $Rosa26^{CreERT2/+}$ and $Krt14^{CreERT2}$ mice (from four independent experiments). $Card14^{LSL-E138A/LSL-E138A}$ $Rosa26^{CreERT2/+}$ mice: isotype (from four independent experiments), anti-TNF (from two independent experiments), anti-GR1 (from two independent experiments). (C) Numbers of S100A9$^+$ myeloid cells were calculated from stained sections of kidney, liver, and lung tissue taken from isotype, anti-TNF or anti-GR1 treated $Card14^{LSL-E138A/LSL-E138A}$ $Rosa26^{CreERT2/+}$ mice and $Card14^{+/+}$ $Rosa26^{CreERT2/+}$ (not antibody treated) controls. Data pooled from three independent experiments. (D) qRT-PCR was performed on d5 ear and lung tissue from $Card14^{+/+}Rosa26^{CreERT2/+}$ control mice and $Card14^{LSL-E138A/LSL-E138A}$ $Rosa26^{CreERT2/+}$ mice treated with either isotype control IgG or anti-TNF. Fold changes were calculated by comparison with the $Card14^{LSL-E138A/LSL-E138A}$ $Rosa26^{CreERT2/+}$ isotype treated group One representative experiment of 2 is shown. (E) Serum quantification of biochemistry for indicators of liver (ALT: alanine aminotransferase and GLDH: glutamate dehydrogenase) and kidney (inorganic phosphate) damage. Sera were collected from nine independent experiments. Differences between groups analysed by one-way ANOVA (A–E). *, p<0.05; **, p<0.01; ***, p<0.001, ****, p<0.0001.

CARD14$^{E138A}$ induction significantly reduced weight loss in the $Card14^{LSL-E138A/LSL-E138A}$ $Rosa26$-$^{CreERT2/+}$ mice, although not to the same extent as anti-TNF treatment (**Figure 8A**). However, at d5, anti-GR1 treated mice were completely protected from hypothermia (**Figure 8B**) and had significantly decreased S100A9$^+$ myeloid cell infiltration into kidney, liver and lung (**Figure 8C**).

Biochemical analyses revealed that treating homozygous mice with either anti-TNF or anti-GR1, normalised ALT, GLDH and inorganic phosphate levels in the serum (**Figure 8E**). This suggested that these blocking antibody treatments protected homozygous mice from organ dysfunction.

Together these results indicated that in $Card14^{LSL-E138A/LSL-E138A}$ $Rosa26^{CreERT2/+}$ mice, CAR-D14$^{E138A}$ signalling in keratinocytes induced systemic release of TNF, which promoted recruitment of myeloid cells into vital organs via increased expression of chemokines. Myeloid cells in combination with upregulated expression of pro-inflammatory genes led to organ dysfunction and severe disease.

## Discussion

Gain-of-function *CARD14* mutations are linked to the development of plaque psoriasis and generalised pustular psoriasis (GPP). *CARD14* variants are also associated with pityriasis rubra pilaris, a rare papulosquamous disorder phenotypically related to psoriasis (*Israel and Mellett, 2018*). In the present study, a conditional knock-in *Card14*$^{E138A}$ mouse strain was used to determine how the highly penetrant psoriasis-associated *CARD14*$^{E138A}$ mutation induces inflammation in adult mice. The effects of CARD14$^{E138A}$ signalling were found to be dose-dependent. Tamoxifen induction of CAR-D14$^{E138A}$ in heterozygous $Card14^{LSL-E138A/+}Rosa26^{CreERT2}$ mice induced skin inflammation phenotypically similar to psoriasis, while homozygous $Card14^{LSL-E138A/LSL-E138A}$ $Rosa26^{CreERT2/+}$ mice developed skin inflammation and a severe systemic disease. Both phenotypes were driven by CAR-D14$^{E138A}$ signalling in keratinocytes and dependent on TNF.

Earlier immunohistochemical studies showed that CARD14 protein was expressed in the basal layer in human skin (*Jordan et al., 2012b*). In contrast, our analyses of mouse skin found that CARD14 was expressed at low levels in undifferentiated keratinocytes, and its expression increased with differentiation both at the RNA and protein level. These results suggest that the localisation of CARD14 expression differs between mouse and human skin. However, the present study also found that CARD14 protein expression increased in human keratinocytes differentiated in vitro. Furthermore, a similar trend was found in publicly available RNA sequencing datasets in which human keratinocytes were differentiated in vitro (GSE73305) and in organotypic epidermal tissue (GSE52954). It will be important to determine whether CARD14 protein levels similarly increase as keratinocytes differentiate in human skin. Nevertheless, regardless of apparent differences in CARD14 distribution between mouse and human, the predominant localisation of CARD14 in the skin is consistent with a function in sensing harmful stimuli, such as pathogens, at the interface between the organism and the environment.

Psoriasis-associated *CARD14* mutations are not linked with inflammatory gut phenotypes, although CARD14 is also expressed at high levels in the gastro-intestinal tract in humans (*Fuchs-Telem et al., 2012*). RNAscope analyses showed that CARD14 is expressed in intestinal epithelial cells (IECs). Consistent with this, tamoxifen induction of CARD14$^{E138A}$ in $Card14^{LSL-E138A/+}Rosa26$-$^{CreERT2}$ mice promotes the expression of pro-inflammatory genes and recruitment of S100A9$^+$

myeloid cells in the colon. However, CARD14$^{E138A}$ expression in the colon was not sufficient to cause any gut tissue damage. These results are similar to the effects of expressing a constitutively active IKK2 mutant in IECs in IKK2[EE]$^{IEC}$ mice, which also induces expression of pro-inflammatory cytokines in the gut and inflammatory cell infiltration without causing overt tissue damage (*Guma et al., 2011*). IKK2[EE]$^{IEC}$ mice, however, develop destructive acute inflammation, with intestinal barrier disruption and bacterial translocation following dextran sulfate sodium (DSS) or lipopolysaccharide (LPS) administration (*Guma et al., 2011*; *Vlantis et al., 2011*). Thus, additional insults might be required for CARD14$^{E138A}$ signalling to induce a gut pathology.

Psoriasis can present as a mild stable disease with sharply demarcated psoriatic plaques in the skin at the prototypical body sites (e.g. elbows, knees, scalp) that predominantly involves the adaptive immune system. Alternatively, psoriasis can be an active severe disease with extensive body surface coverage, involving elevated levels of cytokines and inflammatory markers in the blood and systemic inflammation (*Christophers and van de Kerkhof, 2019*; *Schön, 2019*). This is mainly driven by cells of the innate immune system (*Liang et al., 2017a*). Acute induction of CARD14$^{E138A}$ in *Card14*$^{LSL-E138A/+}$*Rosa26*$^{CreERT2/+}$ mice induced widespread skin lesions in back skin and ears that lacked defined borders. This primarily involved infiltration of innate immune cells into the skin and did not require adaptive immune cells. Prolonged CARD14$^{E138A}$ signalling in *Card14*$^{LSL-E138A/+}$*Rosa26*$^{CreERT2/+}$ mice promoted the development of clearly delineated plaques on the ears with reduced acanthosis in back skin, suggesting less severe skin disease overall. These changes correlated with decreased skin expression of pro-inflammatory genes, decreased serum levels of key pro-inflammatory cytokines and more extensive infiltration of T cells into the skin. Acute expression of CARD14$^{E138A}$ in *Card14*$^{LSL-E138A/+}$*Rosa26*$^{CreERT2}$ and *Card14*$^{LSL-E138A/LSL-E138A}$ *Rosa26*$^{CreERT2/+}$ mice, therefore, appeared to model aspects of active psoriasis. In contrast, chronic CARD14$^{E138A}$ signalling in *Card14*$^{LSL-E138A/+}$*Rosa26*$^{CreERT2}$ mice resulted in a phenotype more similar to mild, stable disease in psoriatic patients. Further research using this model will help to elucidate what factors mediate the transition from acute to chronic lesion.

The fold changes of the majority of proinflammatory genes analyzed at d5 in *Card14*$^{LSL-E138A/+}$*Krt14*$^{CreERT2/+}$ with respect to controls were lower than the fold changes of the *Card14*$^{LSL-E138A/+}$*Rosa26*$^{CreERT2/+}$ mice. The *Krt14*$^{CreERT2}$ driver can induce 'leaky' expression of Cre (*Vasioukhin et al., 1999*) consistent with some *Card14*$^{E138A/+}$ *Krt14*$^{CreERT2/+}$ mice presenting with an inflammatory skin phenotype without tamoxifen injection. Although such mice were excluded from our analyses, it is possible that low concentrations of active Cre in uninduced mice resulted in the expression of limiting amounts of CARD14$^{E138A}$ which chronically activated NF-κB without promoting overt skin inflammation. This may have reduced the fold changes after tamoxifen injection due to negative feedback regulation as seen at 1 m post-tamoxifen injection in *Card14*$^{LSL-E138A/+}$ *Rosa26*$^{CreERT2/+}$ mice. It is also possible that reduced tamoxifen-induction of skin gene expression in *Card14*$^{E138A/+}$ *Krt14*$^{CreERT2/+}$ mice was due to differences to different expression levels of Cre protein between the two Cre-drivers, which use distinct promoters. Regardless, the induced genes were the same in both Cre drivers and the phenotype of both het and hom mice in the *Krt14*$^{CreERT2/+}$ background was similar to the *Rosa26*$^{CreERT2/+}$ histologically and systemically demonstrating that CARD14$^{E138A}$ signalling in keratinocytes alone is sufficient to induce skin and systemic inflammation.

Earlier studies have described the phenotypes of knock-in mice constitutively expressing activated CARD14 variants (*Mellett et al., 2018*; *Sundberg et al., 2019*; *Wang et al., 2018*). Heterozygous *Card14*$^{ΔE138/+}$, *Card14*$^{E138A/+}$ and *Card14*$^{ΔQ136/+}$ mice develop skin inflammation similar to heterozygous *Card14*$^{LSL-E138A/+}$*Rosa26*$^{CreERT2/+}$ mice following tamoxifen induction. In both *Card14*$^{ΔE138/+}$ and *Card14*$^{ΔQ136/+}$ mice, blocking the IL-17A/IL-23 axis ameliorates skin inflammation. Anti-IL-17A also reduced the acanthosis that develops in the back skin of *Card14*$^{LSL-E138A/+}$*Rosa26*$^{CreERT2/+}$ mice 5d after tamoxifen induction, but did not reduce acanthosis in ears or profoundly alter expression of pro-inflammatory genes. Acute induction of skin inflammation by CARD14$^{E138A}$, therefore, is relatively independent of IL-17A and rather driven by overproduction of TNF by keratinocytes. Furthermore, inducible expression of CARD14$^{E138A}$ promoted the acute and chronic development of skin inflammation independently of adaptive immune cells (and T cell production of IL-17A). It will be important to test whether anti-TNF treatment could prevent chronic skin inflammation induced by CARD14$^{E138A}$.

The absence of a requirement for adaptive immune cells for the development of skin inflammation in *Card14*$^{LSL-E138A/+}$*Rosa26*$^{CreERT2/+}$ mice contrasts with a published study by Wang et al. where

the inflammatory skin phenotype in mice constitutively expressing $CARD14^{\Delta Q136}$ was reduced by RAG1 deficiency. However, our data resemble other murine models of skin inflammation originating from constitutive signalling in keratinocytes where the absence of adaptive immune cells does not ameliorate skin disease and can even exacerbate it (*Leite Dantas et al., 2016*; *Uluçkan et al., 2019*). Accordingly, in the absence of Tregs, innate immune cells play a more significant role in the imiquimod-induced psoriasis-like skin pathology (*Hartwig et al., 2018*; *Stockenhuber et al., 2018*). $Card14^{\Delta Q136/+}$ mice only displayed a histopathologic phenotype in tail skin, but not ear skin. It is possible that the adaptive immune system plays different roles in these different anatomical regions. Alternatively, the inflammatory skin phenotype that develops in $Card14^{\Delta Q136/+}$ mice may involve developmental alterations that occur in utero that require adaptive immune cells.

Systemic disease was not described in mice constitutively expressing the $CARD14^{E138A}$ variants (*Mellett et al., 2018*; *Sundberg et al., 2019*; *Wang et al., 2018*), with all analyses focusing only on inflammatory skin phenotypes . However, heterozygous $Card14^{E138A/+}$ mice have reduced survival after birth, while $Card14^{\Delta E138A/\Delta E138A}$ mice die in utero or shortly following birth. It is possible that variant CARD14 signalling during early life induces a severe, systemic disease in these strains that decreases survival. In addition, in utero expression of variant CARD14 could mean that, by the time of analysis of surviving adult mice, the disease state may have already developed from severe/active to mild/stable. In line with this, the inflammatory back skin phenotype, which coincided with severe disease in $Card14^{LSL-E138A/LSL-E138A}$ $Rosa26^{CreERT2/+}$ mice, was not evident in constitutive $Card14^{\Delta Q136/+}$ mice, and only briefly detected (between 5 and 7 days of age) in constitutive $Card14^{\Delta E138A/+}$ mice (*Wang et al., 2018*).

Our analyses of $Card14^{LSL-E138A/+}Rosa26^{CreERT2}$ mice and $Card14^{LSL-E138A/LSL-E138A}$ $Rosa26\text{-}^{CreERT2/+}$ mice revealed a $Card14^{E138A}$ gene dosage effect, with increased $CARD14^{E138A}$ expression inducing a more severe acute phenotype. This presumably explains why *CARD14* psoriasis-associated mutations are generally heterozygous. Weaker NF-κB activating variants, such as $CARD14^{G117S}$, have been found in their homozygous forms in children with severe psoriasis suggesting that the overall level of NF-κB activation is important (*Craiglow et al., 2018*; *Israel and Mellett, 2018*; *Sugiura et al., 2015*). Genetic background also contributes to the penetrance of gain-of-function *CARD14* mutations in promoting psoriasis (*Bhalerao and Bowcock, 1998*), as recently confirmed in mouse studies in which the survival of mice with a constitutive $Card14^{E138A}$ mutation is increased by crossing C57BL/6 mice onto a 129S4 background (*Sundberg et al., 2019*).

The severe, systemic disease that developed in homozygous $Card14^{LSL-E138A/LSL-E138A}$ $Rosa26\text{-}^{CreERT2/+}$ mice following tamoxifen induction had similarities with exacerbations in GPP patients. These included skin disease that covered the majority of the body surface and hypothermia, the mouse equivalent of human fever (*Gordon, 2005*). The extensive scaling of mutant mouse lips also appeared to resemble cheilitis, a common manifestation of GPP (*Ly et al., 2019*). Although pustules were not macroscopically visible (perhaps reflecting differences between human and mouse skin), histological analysis revealed extensive pustule formation within the epidermis and stratum corneum following $CARD14^{E138A}$ expression. $Card14^{LSL-E138A/LSL-E138A}$ $Rosa26^{CreERT2/+}$ mice also presented with myeloid cell infiltration in multiple vital organs and evidence of multi-organ dysfunction, similar to patients with GPP. The skin transcriptome of $Card14^{LSL-E138A/+}Rosa26^{CreERT2}$ mice also shared similarities with the skin transcriptome of the GPP patient expressing the $CARD14^{E138A}$ variant. The increased expression of *Steap1* and *Steap4* mRNAs in the skin of $Card14^{LSL-E138A/+}Rosa26^{CreERT2}$ mice after tamoxifen injection is particularly interesting since these genes have been found to be upregulated in several types of pustular psoriasis and are known to be important in neutrophil chemotaxis (*Liang et al., 2017b*). To our knowledge, $Card14^{LSL-E138A/LSL-E138A}$ $Rosa26^{CreERT2/+}$ mice are the first to mimic the phenotypes of GPP and might be useful in future studies to understand more about the etiopathogenesis of this severe form of psoriasis.

Both GR1 antibody (to deplete neutrophils and monocytes) and TNF antibody significantly reduced disease severity in $Card14^{LSL-E138A/LSL-E138A}$ $Rosa26^{CreERT2/+}$ mice following tamoxifen administration. Current treatments for GPP patients include granulocyte/monocyte apheresis and anti-TNF (*Fujisawa et al., 2015*; *Koike et al., 2017*) suggesting that $Card14^{LSL-E138A/LSL-E138A}Rosa26\text{-}^{CreERT2/+}$ mice could be used for pre-clinical testing of new medicines for GPP treatment. The present study indicates that constitutive $CARD14^{E138A}$ activation of NF-κB in keratinocytes is sufficient to promote skin inflammation and systemic inflammatory disease. This raises the possibility that it may

be feasible to develop topical medicines that block CARD14 signalling in the skin to treat CARD14-induced psoriasis and systemic inflammation.

# Materials and methods

## Key resources table

| Reagent type (species) or resource | Designation | Source or reference | Identifiers | Additional information |
|---|---|---|---|---|
| Genetic Reagent (*M. musculus*) | Card14<sup>LSL-E138A</sup> | Taconic | MGI:6111507 | Mice were bred into the C57BL6/J background for > 8 generations by the Ley lab. Strain name at the Francis Crick Institute SLAT. |
| Genetic Reagent (*M. musculus*) | Rosa26<sup>CreERT2</sup> | PMID:12582257 | RRID:IMSR_TAC:10471 | Strain name at the Francis Crick Institute BRAW |
| Genetic Reagent (*M. musculus*) | Villin<sup>CreERT2</sup> | PMID:15282745 | RRID:IMSR_JAX:020282 | Strain name at the Francis Crick Institute BRGU |
| Genetic Reagent (*M. musculus*) | Krt14<sup>CreERT2</sup> | PMID:14742263 | (MGI:4357971) | Mice were bred into the C57BL6/J background for > 8 (more like N5 for the SLDD12) generations by the Ley lab Strain name at the Francis Crick Institute SLBN |
| Genetic Reagent (*M. musculus*) | Rag1<sup>-/-</sup> | PMID:7926785 | (MGI:2448994) | Historically Backcrossed 12 x to C57BL/6J total N unknown Strain name at the Francis Crick Institute BRAU |
| Gene (*M. musculus*) | Card14 | *Mus musculus* Mouse Genome Informatics | MGI:2386258 | |
| Cell line (*H. sapiens*) | NHEK | Lonza | Cat #00192627 | |
| Antibody | IgG1 isotype control (mouse monoclonal) | BioXcell | MOPC-21 | 0.5 mg per injection |
| Antibody | Anti-Il17a (mouse monoclonal) | BioXcell | clone 17F3 | 0.5 mg per injection |

*Continued on next page*

*Continued*

| Reagent type (species) or resource | Designation | Source or reference | Identifiers | Additional information |
|---|---|---|---|---|
| Antibody | IgG2b isotype control (Rat monoclonal) | BioXcell | LTF-2 | 0.5 mg per injection |
| Antibody | Anti-Gr1 (Rat monoclonal) | BioXcell | clone RB6-8C5 | 0.5 mg per injection |
| Antibody | rat IgG1 isotype control (Rat monoclonal) | BioXcell | TNP6A7 | 0.5 mg per injection |
| Antibody | Anti-TNFa (Rat monoclonal) | BioXcell | clone XT3.11 | 0.5 mg per injection |
| Antibody | Anti-Ki67 (Rabbit monoclonal) | Abcam | ab16667 | 1/350 |
| Antibody | Anti-Involucrin (Rabbit monoclonal) | In house | ERL-3 | Produced by the Crick Cell Services. 1/800 |
| Antibody | Anti-S100a9 (Rat monoclonal) | In house | 2b10 | 1/1000 (Can be purchased from abcam ab105472) |
| Antibody | Anti-Endomucin (Rat monoclonal) | Santa Cruz | sc-65495 | 1/400 |
| Antibody | anti-FLAG (Mouse monoclonal) | Sigma | F1804 | 1/1000 |
| Antibody | anti-CARD14 (Rabbit polyclonal) | This paper | CUK-1813 | Produced by Covalab 1/1000 |
| Antibody | anti-Hsp90 (Rabbit polyclonal) | Santa Cruz | sc-7947 | 1/5000 |
| Sequence-based reagent | *Card14* oligo 1 | This paper | PCR oligo | TCAACATTATCT TCCAAGCTCC |
| Sequence-based reagent | *Card14* oligo 2 | This paper | PCR oligo | TGACCTCACGT TTCATGCG |
| Commercial assay or kit | SuperScript VILO cDNA Synthesis Kit | Life Technologies | 11754250 | |
| Commercial assay or kit | RNeasy mini kit | Qiagen | 74106 | |
| Commercial assay or kit | LEGENDplex, | Biolegend | 740150 | |
| Commercial assay or kit | TaqMan Gene Expression Master Mix | Thermo Fisher | 4369514 | |
| Commercial assay or kit | *Card14* RNAscope | ACDBio | Probe - Mm-Card14 Cat No 476041 | |
| Chemical compound, drug | Corn oil | Sigma | C8267 | 100 ul per injection |

*Continued on next page*

*Continued*

| Reagent type (species) or resource | Designation | Source or reference | Identifiers | Additional information |
|---|---|---|---|---|
| Chemical compound, drug | Tamoxifen | Sigma | T5648 | 2 mg per injection |
| Software, algorithm | Image J | NIH, Bethesda, MD | RRID:SCR_003070 | https://imagej.nih.gov/ij/ |
| Software, algorithm | GraphPad Prism | GraphPad Prism (https://graphpad.com) | RRID:SCR_002798 | GraphPad Prism eight software for Mac |

## Animals

Mice were bred and maintained under specific pathogen–free conditions at the Francis Crick Institute. Experiments were performed in accordance with UK Home Office regulations and endorsed by the Francis Crick Institute Animal Welfare and Ethical Review Body under the Procedure Project Licence 70/8819. *Rosa26*^CreERT2^ (*Seibler et al., 2003*), *Krt14*^CreERT2^ (*Hong et al., 2004*), *Villin*^CreERT2^ (*el Marjou et al., 2004*) and *Rag1*^-/-^ (*Spanopoulou et al., 1994*) mouse lines have been described previously.

The *Card14*^LSL-E138A^ mouse strain was made by Taconic (official strain name *Card14*^tm1.1Abow^). The targeting strategy allows the generation of a constitutive Knock-In (KI) of a 3xFLAG tag or a conditional Knock-In of a point mutation (KI-PM) in the *Card14* gene. The 3' part of intron 3, containing the splice acceptor site, and exons 4 to 22, including the 3'-untranslated region, (UTR) were flanked by loxP sites and inserted into intron 3 of the genomic *Card14* locus (*Figure 1—figure supplement 1B*). An additional polyadenylation signal (hGHpA: human growth hormone polyadenylation signal) was inserted between the 3' UTR and the distal loxP site to prevent transcriptional read-through. The sequence for the 3xFLAG tag with linker (GGGS-DYKDHDGDYKDHDIDYKDDDDK) was inserted between the last amino acid and the translation termination codon in exon 22 of the wildtype *Card14* cDNA (fused to exons 4 to 22). The insertion allows the translation of a wildtype CARD14-3xFLAG tag fusion protein. The E138A mutation (GAG >GCG) was introduced into exon 5 downstream of the distal loxP site. Positive selection markers were flanked by FRT (neomycin resistance - NeoR) and F3 (puromycin resistance - PuroR) sites and inserted downstream of the proximal loxP site and upstream of the distal loxP site. The targeting vector was generated using BAC clones from the C57BL/6J RPCIB-731 BAC library and was transfected into the Taconic Biosciences C57BL/6N Tac ES cell line.

Transgenic mice containing the construct were crossed to a Flp deleter strain to remove the neo selection cassette. The resulting *Card14*^LSL-E138A/LSL-E138A^ mice expressed the wild type CARD14-3xFLAG tag fusion protein from the endogenous *Card14* promoter and thus recapitulated the expression pattern of the *Card14* gene. These mice were crossed to the indicated inducible Cre driver strains. After tamoxifen-induced Cre-mediated recombination, the minigene was excised, resulting in expression of *Card14*^E138A^. Mouse genotyping was carried out by TransNetyx.

To verify tissue-specific Cre deletion, DNA extracted from ear snips or colon was used for genotyping by PCR (KAPA Taq PCR Kit). The fragment amplified with oligos 1 (TCAACATTATCTTCCAAGCTCC) and 2 (TGACCTCACGTTTCATGCG) detected the WT *Card14* locus (1818 bp), the Knock-In allele (1310 bp), and the mutant *Card14*^E138A^ allele after Cre deletion (1899bp).

## In vivo procedures

To induce expression of CARD14^E138A^, *Card14*^LSL-E138A/+^ mice, crossed to the appropriate CreERT2 driver strain, were given intraperitoneal (I.P.) injections of tamoxifen (Sigma) dissolved in corn oil (Sigma). Each mouse received a total of three I.P. injections (2 mg/injection) of tamoxifen over three consecutive days.

For all blocking antibody experiments, mice received I.P. injections of antibody on days −1, 1 and 3. In all cases, I.P. tamoxifen was given on days 0, 1 and 2. (On day 1, when mice received two I.P. injections, the blocking antibody was given in the morning and tamoxifen in the afternoon). Each

antibody injection contained 0.5 mg of blocking antibody or isotype control. All antibodies were purchased from BioXcell. Mice in each experimental cohort were treated with either a blocking antibody or the appropriate isotype control. Antibody/isotype control combinations were as follows: TNF antibody (clone XT3.11) and rat IgG1 isotype control (TNP6A7); GR1 antibody (anti-Ly6G/Ly6C, clone RB6-8C5) and rat IgG2b isotype control (clone LTF-2); IL-17A antibody (clone 17F3) and mouse IgG1 isotype control (MOPC-21).

## Flow cytometry

Flow cytometric analyses were performed with 5 million splenocytes, 5 million cells from the lamina propria of the colon, 5 million cells from the epithelium of the colon, 4x skin punches (total surface area of approximately 200 mm$^2$), or 1 ear.

Skin was shaved and excess fat scraped away from the dermis. Ears were split into dorsal and ventral halves to aid digestion. Skin and ears were finely minced. Each skin or ear sample was incubated (1 hr, 37°C, with vigorous shaking) in digestion cocktail (RPMI containing 0.4 mg/mL liberase TL (Roche), 1 mg/mL collagenase D (Roche), 50 ug/mL DNase I (Roche)).

Individual, whole colons were flushed with ice-cold PBS, cut into 0.5–1 cm long pieces, and washed thoroughly by vortexing in PBS. The epithelial cell layer was removed and collected by 2x incubation of tissue pieces in pre-digestion solution (1x HBSS, calcium and magnesium free, containing 10 mM HEPES, 5 mM EDTA, 2 mM DTT, 1% FCS, 1x penicillin-streptomycin) at 37°C, with vigorous shaking, for 20 min followed by sample straining and collection of the flow-through. Tissues pieces were then washed in HBSS with 5% FCS for 10 min at 37°C, with vigorous shaking, followed by further straining and collection of flow-through. Pooling of collected supernatants from the 2 pre-digestion steps and the washing step produced the colon epithelium sample (stained and analysed as below). Remaining tissue pieces (consisting of the colon lamina propria) were then minced and incubated in digestion solution (1x HBSS, with calcium and magnesium, containing 10 mM HEPES, 1% FCS, 1x penicillin-streptomycin, 50 ug/mL DNase I, 0.4 mg/mL Liberase TL) at 37°C, with vigorous shaking, for 30 min.

Digested skin, ear, colon lamina propria samples, and whole spleens were filtered through a 100 µm cell strainer to produce single cell suspensions. Cells were incubated in FcR block (anti-CD16/CD32, clone 93, Biolegend, 0.5 mg/mL) for 20 min at 4°C, followed by washing in PBS containing 0.1% BSA. Samples were then stained in saturating concentrations of antibodies diluted in PBS/BSA for 1 hr at 4°C, followed by washing in PBS/BSA. Samples were fixed in 4% paraformaldehyde (IC fixation buffer, eBioscience) for 15 min, followed by washing in PBS/BSA. Counting beads (CountBright Absolute counting beads, ThermoFisher Scientific) were added to samples prior to analysis on a BD LSR Fortessa X20. Compensation was performed with the aid of beads (Ultracomp eBeads, ThermoFisher Scientific). Data analysis was performed with FlowJo V10.5.3 software (Tree Star).

CD4 BV650 (clone RM4-5), CD8 BV510 (clone 53–6.7), CD11b BV605 (clone M1/70), CD11c PerCPcy5.5 (clone N418), CD25 APC (clone PC61), CD64 FITC (clone X54-5/7.1), MHC-II BV510 (clone M5/114.15.2), and TCRβ PerCPcy5.5 (clone H57-597) were purchased from Biolegend. CD19 APCcy7 (clone 1D3) and Ly6G PE (clone 1A8) were purchased from BD Pharmingen. CD45.2 APCef780 (clone 104), CD45.2 ef450 (clone 104), MerTK PEcy7 (clone DS5MMER), NK1.1 PEcy7 (clone PK136), and TCRγ/δ PE (clone eBioGL3) were purchased from eBioscience. Live/dead fixable blue dead cell stain was purchased from ThermoFisher Scientific.

For flow cytometric gating (*Figure 1—figure supplement 3*), all populations were first gated on: single, live and CD45$^+$ cells. Gating strategies for individual cell populations were then as follows: neutrophils: MHC-II$^-$, CD11b$^+$, Ly6G$^+$; macrophages: Ly6G, MHCII$^+$ and/or CD11b$^+$, Ly6C+ and/or CD64$^+$, MerTK$^+$ CD64$^{hi}$; dendritic cells (DCs): MHC-II$^+$, CD11b$^+$, CD11c$^{hi}$; αβ T cells: TCRβ$^+$; CD4 T cells: TCRβ$^+$, CD4$^+$; CD8 T cells: TCRβ$^+$, CD8$^+$; Tregs: TCRβ$^+$, CD4$^+$, CD25$^+$; B cells: CD19$^+$; γδ T cells: TCRγδ$^+$; NK cells: NK1.1$^+$.

## Immunoplex array and serum biochemistry

Blood was collected by cardiac puncture of dead mice. Blood was allowed to clot and serum separated by centrifugation. Cytokine and chemokine concentrations within serum samples were analysed by immunoplex array (LEGENDplex, Mouse Inflammation Panel, Biolegend) on a BD LSR Fortessa X20, following manufacturer's instructions.

Serum biochemistry was quantified by IDEXX bioanalytics.

## Keratinocyte cultures

Primary mouse keratinocyte cultures were prepared from mouse tail skin. Tails were cut longitudinally and bone removed. After cutting into 3 segments, the pieces were floated in cold 5 mg/ml Dispase (Thermo Fisher, 17105–041) for 4 hr at 4°C. The epidermis was then separated with forceps, and washed in cold PBS twice. Epidermis was floated in TrypLE select (Thermo fisher, 12563011) for 10 min at 37°C to detach the keratinocytes. After TrypLE incubation, epidermis was further minced with forceps and aspirated through a pipette 20 times. The suspended keratinocytes were then transfered into a 50 ml falcon with a filter on the top (100 µm) and centrifuged to pellet the cells. Keratinocytes were cultured in 7 µg/ml Collagen 1 coated plates (Thermo Fisher, A1048301) with CNT-57 medium for 2 days and then changed to CNT-02 for 5d (CELLnTEC, Bern, Switzerland). The medium was changed every second day. Normal human epidermal keratinocytes (NHEK) were purchased from Lonza and cultured in CNT-57 medium.

For differentiation studies, cells were seeded in six-well plates (Thermo Fisher) and grown to pre-confluence, then 2 mM $CaCl_2$ was added and lysates were obtained at the indicated culture times.

## Histopathology and immunohistochemistry

Excised Tissues were fixed in 10% neutral buffered formalin for 48 hr at RT, then stored in 70% ethanol. Tissues were embedded in paraffin and sections (5 µm) were stained with hematoxylin and eosin (H and E). Pictures were taken with a Nikon Eclipse 90i microscope. For acanthosis (epidermal thickness) quantification, 3 measurements from the basement membrane to the stratum corneum were taken for each 10x magnification picture using Image J (NIH, Bethesda). For each mouse, 3 pictures were taken.

For immunohistochemistry, tissue sections were dewaxed three times in xylene for 1 min. Slides were then immersed in a series of decreasing concentrations of ethanol (1 min incubations) ending with distilled water. Heat-mediated antigen retrieval was conducted for Ki67 (Abcam ab16667) and Involucrin (in house) staining. Briefly, slides were immersed in a preheated solution of pH 6 Citrate buffer and further heated for 15 min in a microwave. For S100A9 (2b10 clone; in house) staining, antigen retrieval involved incubation of slides in 0.1% Trypsin solution for 15 min at 37°C. After antigen retrieval, endogenous peroxidases were blocked in 1.6% $H_2O_2$ in PBS for 10 min followed by distilled water for 5 min. The corresponding normal serum diluted to 10% in 1%BSA/PBS was applied for 10 min, at RT. After blocking, anti-KI67 (1/350), anti-involucrin (1/800), anti-S100A9 (1/1000) or anti-Endomucin (Santa Cruz, sc-65495 1/400) was added to slides, which were incubated overnight at 4°C. Antibody excess was washed from slides by three PBS immersions. The corresponding HRP-linked secondary antibody was applied for 45 min at RT and then washed 3x with PBS. Avidin-biotin complex solution was applied for 30 min at RT and washed 3x with PBS. DAB solution was used to develop the signal and the reaction terminated with distilled water. To count dermal infiltrating cells (S100A9) or epidermal Ki67+ cells, three pictures per sample were randomly taken at 10x magnification, the number of infiltrating cells or Ki67[+] cells were counted and the average between the three pictures calculated.

For *Card14* mRNA detection in tissue, RNAscope Probe - Mm-Card14 Cat No 476041 was used following manufacturer's instructions (ACDBio). For *Tnfa* mRNA detection in tissue, RNAscope Probe- Mm-TNFa Cat No. 311081 was used. For localisation of *Tnfa* with different cell types, the following antibodies were used: CK5 (Abcam, ab52635 at 1/6000) for keratinocytes; CD3 (Abcam, ab134096 at 1/750) for T cells and Ly6g (BD Pharmigen, 551459 at 1/500) for neutrophils. RNAscope staining was performed first, followed by IHC. Bound antibodies were visualised using Blue chromogen (Vector, SK-5300) for 3–15 min at RT.

## Protein lysates and western blot

Mouse tissues were lysed in radioimmunoprecipitation assay (RIPA) buffer (50 mM Tris-HCl pH 7.5, 150 mM NaCl, 2 mM EDTA, 1 mM sodium pyrophosphate, 50 mM sodium fluoride, 1 mM sodium vanadate, 0.1% SDS, 1% Triton-X, 0.5% deoxycholate and protease inhibitor cocktail (Roche 11836170001)). Protein levels were quantified (Pierce BCA protein assay, Thermo Scientific) and normalised. Lysates were boiled in Laemmli buffer, resolved by SDS-PAGE and transfered to a PVDF

membrane (Trans-Blot Turbo Transfer Packs, Biorad). Membranes were blocked for 1 hr at RT with 5% skimmed milk and incubated with anti-FLAG M2 (Sigma F1804); or anti-Hsp90 (Santa Cruz; sc-7947). Bound HRP-labelled secondary antibody (anti-mouse IgG; Southern Biotech) was visualised by enhanced chemiluminescence (ECL, Millipore).

## RNA extraction and quantification by RT-qPCR

Tissues were harvested and immersed in RNAlater (Thermofisher) for 48 hr at 4°C and then stored at −80°C. To extract RNA, the tissues were blended in the presence of trizol with IKA T25 digital Ultra-turrax tissue homogeniser. Chloroform was added and mixed vigorously. Following 15 min centrifugation at maximum speed at 4°C the aqueous phase was transferred to a RNeasy mini kit (Qiagen #74106) to purify RNA, following manufacturer's instructions. Quantity and purity of RNA were analysed using a NanoDrop ND-1000 Spectrophotometer.

SuperScript VILO cDNA Synthesis Kit (Life Technologies #11754250) was used to obtain cDNA from 1 µg of RNA. For RT-qPCR, QuantStudio 5 Real-Time PCR was used according to the instructions provided by the manufacturers. Induction of genes was calculated with the delta delta CT method, using *Hprt* as a housekeeping gene. Real-time quantitative PCR (qPCR) was performed using TaqMan Gene Expression Master Mix and predesigned probes (Thermo Fisher Scientific, Waltham, MA) for the following mouse genes: *Card14* (Mm00459947_m1), *S100a9* (Mm00656925_m1), *Tnf* (Mm00443258_m1), *Il1f9* (Mm00463327_m1), *Il17c* (Mm00521397_m1), *Hprt* (Mm03024075_m1), *Il22* (Mm01226722_g1), *Il23a* (Mm01160011_g1), *Il17a* (Mm00439618_m1), *Ccl20* (Mm01268754_m1), *Cxcl2* (Mm00436450_m1), *Cxcl3* (Mm01701838_m1), *Nos2* (Mm00440485_m1). For human samples, *CARD14* (Hs01106904_m1), *18S* (Hs99999901_s1) and *IVL* (Hs00846307_s1) probes were used.

## RNA-Seq and data analysis

RNA was isolated from tissue as detailed above. RNA-sequencing was performed by the Crick advanced sequencing facility using the polyA KAPA mRNA Hyper Prep kit (Roche) and coding-mRNA as starting material. Sequencing was performed on an Illumina HiSeq4000 instrument with single end reads of at least 75 bp. The RNA-Seq data generated in this article was deposited in the GEO repository (GSE149880).

The 'Trim Galore!' utility version 0.4.2 was used to remove sequencing adaptors and to quality trim individual reads with the q-parameter set to 20 (https://www.bioinformatics.babraham.ac.uk/projects/trim_galore/). Then sequencing reads were aligned to the mouse genome and transcriptome (Ensembl GRCm38 release-89) using RSEM version 1.3.0 (*Li and Dewey, 2011*) in conjunction with the STAR aligner version 2.5.2 (*Dobin et al., 2013*). Sequencing quality of individual samples was assessed using FASTQC version 0.11.5 (https://www.bioinformatics.babraham.ac.uk/projects/fastqc/) and RNA-SeQC version 1.1.8 (*DeLuca et al., 2012*). Differential gene expression was determined using the R-bioconductor package DESeq2 version 1.14.1 (*Love et al., 2014*) (http://www.R-project.org). Gene set enrichment analysis (GSEA) was conducted as described in *Subramanian et al., 2005*.

## Statistical analysis

Data were analysed using GraphPad Prism 8 software for Mac (GraphPad; La Jolla, CA, USA). To compare the mean values between two groups, unpaired two-tailed Student t-tests was used. Statistical differences in mean values between three or more experimental groups were determined with one-way ANOVA Tukey's multiple comparison test. p values < 0.05 were considered statistically significant. Significance was noted in the graphs as stars: *p<0.05; **p<0.01; ***p<0.001 and ****p<0.0001.

## Acknowledgements

We are grateful to the Francis Crick Institute Flow Cytometry, Advanced Sequencing, Biological Research and Experimental Histopathology Facilities for their help during the production of this work. We also thank members of the Ley Laboratory for their support during this project and Dr Francesca Capon (King's College London, UK) and Professor Ben Seddon (University College London, UK) for their feedback on the manuscript. The advice of Professor Arthur Kaser (University of

Cambridge, UK) is also gratefully acknowledged for his insights on TNF-induced inflammation. We thank the members of the Crick Digital Development Team, particularly Amy Strange, Luke Nightingale, and Jude Pinnock for excellent technical support. AMB and AH were supported by a grant from the National Institutes of Health (USA) (R01AR05026). SCL was supported by a grant from the Francis Crick Institute, which receives its core funding from Cancer Research UK (FC001103), the UK Medical Research Council (FC001103) and the Wellcome Trust (FC001103), and project grants from the National Psoriasis Foundation USA (WMIS_P74088 for JM) and the British Heart Foundation (PG/15/57/31580 for LVW).

## Additional information

### Funding

| Funder | Grant reference number | Author |
| --- | --- | --- |
| Francis Crick Institute | FC001103 | Joan Manils<br>Louise V Webb<br>Julia Janzen<br>Stefan Boeing<br>Steven C Ley |
| National Psoriasis Foundation | WMIS_P74088 | Joan Manils |
| British Heart Foundation | PG/15/57/31580 | Louise V Webb |
| National Institutes of Health | R01AR05026 | Ashleigh Howes<br>Anne M Bowcock |

The funders had no role in study design, data collection and interpretation, or the decision to submit the work for publication.

### Author contributions

Joan Manils, Louise V Webb, Conceptualization, Data curation, Formal analysis, Investigation, Writing - original draft, Writing - review and editing; Ashleigh Howes, Conceptualization, Investigation; Julia Janzen, Resources, Investigation, Project administration; Stefan Boeing, Data curation, Software, Formal analysis; Anne M Bowcock, Conceptualization, Funding acquisition; Steven C Ley, Conceptualization, Formal analysis, Supervision, Funding acquisition, Investigation, Writing - original draft, Writing - review and editing

### Author ORCIDs

Joan Manils (ID) https://orcid.org/0000-0001-8429-1295
Anne M Bowcock (ID) http://orcid.org/0000-0001-8691-9090
Steven C Ley (ID) http://orcid.org/0000-0001-5911-9223

### Ethics

Animal experimentation: Mice were bred and maintained under specific pathogen-free conditions at the Francis Crick 787 Institute. Experiments were performed in accordance with UK Home Office regulations and endorsed by the Francis Crick Institute Animal Welfare and Ethical Review Body under the Procedure Project Licence 70/8819. Rosa26CreERT2 (Seibler et al., 2003), Krt14CreERT2 (Hong et al., 2004), VillinCreERT2 (el Marjou et al., 2004) and Rag1-/- (Spanopoulou et al., 1994) mouse lines have been described previously.

### Decision letter and Author response

Decision letter https://doi.org/10.7554/eLife.56720.sa1
Author response https://doi.org/10.7554/eLife.56720.sa2

## Additional files

### Supplementary files
• Transparent reporting form

### Data availability
The RNA-Seq data generated in this article was deposited in the GEO repository (GSE149880).

The following dataset was generated:

| Author(s) | Year | Dataset title | Dataset URL | Database and Identifier |
|---|---|---|---|---|
| Manils J, Webb L, Boeing S | 2020 | Expression analysis of WT or Card14E138A ears 5 days and 1 month after injecton of tamoxifen. | https://www.ncbi.nlm.nih.gov/geo/query/acc.cgi?acc=GSE149880 | NCBI Gene Expression Omnibus, GSE149880 |

The following previously published datasets were used:

| Author(s) | Year | Dataset title | Dataset URL | Database and Identifier |
|---|---|---|---|---|
| Asare A, Levorse J, Fuchs E | 2017 | A spatio-temporal characterization of the transcriptional landscape of epidermal development | https://www.ncbi.nlm.nih.gov/geo/query/acc.cgi?acc=GSE75931 | NCBI Gene Expression Omnibus, GSE75931 |
| Bin L, Deng L, Yang H, Zhu L, Wang X, Edwards MG, Richers B, Leung DYM | 2016 | RNA-sequencing transcriptome profiling of normal human keratinocytes differentiation | https://www.ncbi.nlm.nih.gov/geo/query/acc.cgi?acc=GSE73305 | NCBI Gene Expression Omnibus, GSE73305 |
| Vanessa L-P, Kun Q, Jiajing Z, Dan EW, Brook CB, Zurab S, Brian JZ, Lisa DB, Rios EJ, Shiying T, Markus K, Paul AK | 2014 | A LncRNA-MAF/MAFB transcription factor network regulates epidermal differentiation | https://www.ncbi.nlm.nih.gov/geo/query/acc.cgi?acc=GSE52954 | NCBI Gene Expression Omnibus, GSE52954 |

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
