## [Decision Letter]

**Acceptance summary:**

In the skin disease psoriasis in humans, a mutation has been found in the CARD14 molecule. In the present paper, a murine model with this mutation *Card14*^E138A^ was developed. The study shows that induced expression of this mutant in the skin drives a psoriasis phenotype in mice. An important finding is that neutralization of TNFα is more effective than neutralization of IL-17A, at least in the acute phase. In addition, the investigations suggest that anti-inflammatory therapies that target CARD14 signaling in keratinocytes, rather than systemic biologicals, might be an effective treatment early in disease progression.

**Decision letter after peer review:**

Thank you for submitting your article "CARD14^E138A^ signalling in keratinocytes induces TNF-dependent skin and systemic inflammation" for consideration by *eLife*. Your article has been reviewed by three peer reviewers, including Carla V Rothlin as the Reviewing Editor and Reviewer #1, and the evaluation has been overseen by Jos van der Meer as the Senior Editor. The following individual involved in review of your submission has agreed to reveal their identity: Ellen Vandenbogaard (Reviewer #3).

The reviewers have discussed the reviews with one another and the Reviewing Editor has drafted this decision to help you prepare a revised submission.

Summary:

The authors developed and characterized an inducible murine model for *Card14*^E138A^, a psoriasis-associated mutation in humans. Importantly, this approach enables them to report that induced expression of this mutant in the skin is sufficient for driving a psoriasis phenotype in mice. They also find that neutralization of TNFα provides a more effective therapeutic effect than neutralization of IL-17A, at least in the acute phase. The authors also generated homozygous mutant mice, which display more severe systemic effects, demonstrating a clear gene dosage effect for this mutation. The study is of significant relevance to the field, well-designed and robustly performed. Below we outline a series of revisions to strengthen or clarify the authors' conclusions.

1) The authors have modeled a human mutation into the mouse *Card14* gene. Can the authors add a comparison on the murine gene and the human ortholog in terms of their homology?

2) The authors defined the expression of CARD14 through western blot and RNAscope approaches. The results are in line with previously published murine data, as well as with single cell sequencing data from the Kaspar lab (http://kasperlab.org/mouseskin). However, the results seem to be in contrast to protein expression profiles reported in human skin, in which CARD14 protein is mostly found in basal cells (of note: Figure 1B also shows signal in basal cells). This requires a thorough analysis or at least a discussion on the differences between mice and human and potential consequences for disease mechanisms.

3) Throughout the manuscript, the authors link their results to 'psoriasis' (e.g. disease markers, therapeutic response). However, the investigated CARD14 mutation is related to GPP, not (or to a lesser extent) to other psoriasis endotypes like PV (Berki et al., 2015). The characterization of the inflammatory response includes several markers (e.g. KI67, S100A9, cytokines and chemokines). But most are not considered specific for psoriasis, and are related to skin inflammation or an activated epidermis in general. Therefore, a more comprehensive analysis including more specific markers for psoriasiform inflammation would be preferable. It would be also ideal to compare these findings with data on these markers in human psoriasis, and comparing GPP vs. PV. This would allow the discrimination of the model phenotype and specification of the model being a true GPP or PV model. For example, RNAseq analysis would allow for such a comparison given the publicly available RNAseq data sets from psoriasis, atopic dermatitis and other inflammatory skin conditions.

4) The authors assess the contribution of adaptive immunity to the phenotypes described by crossing *Card14* mutants to Rag1^-/-^ mice. As no significant differences are detected in the absence of T cells and B cells, the authors conclude that adaptive immunity was not required for acute skin inflammation. This is in keeping with the relatively short time of evaluation after the induction of expression of the mutant *Card14* (5 days). Is adaptive immunity relevant at later stages of the model. If so, is anti-TNF still therapeutically effective?

5) The study included the analysis of different cytokines (Figures 1 and 2). The results raise the question as to whether ubiquitous or KC-specific expression of *Card14*^E138A^ result in a different inflammatory profile. A table listing all the cytokines tested and their outcome in both setting would be beneficial to easily compare and contrast between the two models. Along the same line, it would be ideal to include measurements on the cellular infiltrate in the KC-specific model.

6) There are a number of experiments in which controls are missing. For example, include control *Card14*^+/+^*Rosa26*^CreERT2/+^ in Figure 3C, Figure 3—figure supplement 1B.

7) The data in Figure 6—figure supplement 2D is relevant and would be ideal to be included in a main figure. We also suggest to include this in the main figure and replace the pictures shown in Figure 6C.

8) The finding that systemic effects are mediated through keratinocyte-driven severe skin inflammation is highly interesting and could have clinical impact. This finding deserves to be better highlighted in the manuscript. This may also suggest that patients could benefit, early in disease progression, from local anti-inflammatory therapies specifically targeting the keratinocytes, rather than systemic biologicals.

Revisions expected in follow-up work:

1) We recognize that a more detailed characterization of the model and its comparison with data sets on human psoriasis, GPP and PV would require significant additional experiments and can be performed as a follow up work. A discussion on how such an approach can complement the current findings to discriminate this with other disease models would be ideal.

2) If the authors have not tested the efficacy of anti-TNF beyond the acute period, they should clarify that this would be important to be tested in follow up work.

3) We recognize that a thorough comparative analysis on CARD14 expression between mouse and human samples would require significant additional experiments and can be performed as a follow up work. A discussion on the implications of differential expression is warranted for this report.

---

## [Author Response]

Revisions for this paper:1) The authors have modeled a human mutation into the mouse Card14 gene. Can the authors add a comparison on the murine gene and the human ortholog in terms of their homology?

We have added a sentence at the beginning of the Results section:

“Human and mouse CARD14 proteins are highly homologous, with 77% amino acid identity overall and 80% identity in the coiled-coil region in which the *CARD14*^E138A^ mutation is located (Figure 1—figure supplement 1A).”

Additionally, we have added a human and mouse CARD14 amino acid sequence alignment in of the conserved region around the psoriasis point mutation E138 (Figure 1—figure supplement 1A).

2) The authors defined the expression of CARD14 through western blot and RNAscope approaches. The results are in line with previously published murine data, as well as with single cell sequencing data from the Kaspar lab (http://kasperlab.org/mouseskin). However, the results seem to be in contrast to protein expression profiles reported in human skin, in which CARD14 protein is mostly found in basal cells (of note: Figure 1B also shows signal in basal cells). This requires a thorough analysis or at least a discussion on the differences between mice and human and potential consequences for disease mechanisms.

Two studies have suggested that human CARD14 is expressed in the basal layer of the epidermis based IHC analyses. In Jordan et al., 2012, a Sigma antibody (HPA023388) was used to stain frozen skin sections. The second study (Fuchs-Telem et al., 2012), investigating the role of CARD14 in PRP, showed basal staining apparently using a different CARD14 antibody (NBP1-88598; Novus Biological). However, this antibody actually appears to be identical to the Sigma antibody (same immunogen sequence, same host species and same staining pictures used to advertise the antibody). Both suppliers have discontinued these antibodies and no staining with these reagents are reported for CARD14 in the human protein atlas.

We have tried the Sigma CARD14 antibody ourselves for western blotting (J.M. and S.C.L.). Using lysates from HACAT cells overexpressing CARD14-FLAG, the antibody failed to detect any specific signal although the recombinant CARD14 was clearly detected with anti-FLAG. Jordan et al., 2012 performed a negative control stained with 1% serum in lieu of the primary antibody. While this controls for unspecific retention of primary antibodies by the tissue, it does not control for unspecific binding of the specific CARD14 antibody (a better control for unspecific binding would be CARD14 knock out tissue or antibody adsorption with CARD14 recombinant protein prior to staining). Fuchs-Telem et al., 2012 do not show any negative controls. We are concerned that the basal staining with the commercial CARD14 antibodies do not actually show CARD14 protein levels and do not feel that it is possible at this stage to make a definitive statement about differences in CARD14 expression in the skin of mice and humans.

Nevertheless, similar to mouse keratinocytes, we found that differentiation of human keratinocytes increased levels of CARD14 protein and mRNA (Figure 1—figure supplement 1E and F). This is consistent with published RNA sequencing data with cultured human keratinocytes and organotypic epidermis (Bin et al., 2016) (Lopez-Pajares et al., 2015). It will be important to determine whether CARD14 protein levels similarly increase as keratinocytes differentiate in human skin and to confirm CARD14 protein expression in the basal layer using different CARD14 antibodies with validated specificity.

We have modified the text in the Results describing CARD14 localization to take in to account these considerations:

“RNAscope analysis revealed that *Card14* mRNA expression in healthy mouse skin was restricted to the upper (most differentiated) keratinocytes, the outer root sheath of the hair and sebocytes in the skin (Figure 1B). […] The *CARD14* mRNA expression data agree with published transcriptomic datasets of differentiating human keratinocytes (Bin et al., 2016) and organotypic epidermis (Lopez-Pajares et al., 2015) (Figure 1—figure supplement 1G).”

Additionally, RNAscope staining of *Card14* in the skin of *Card14^LSL-E138A/+^Rosa26^CreERT2/+^* mice 5 days after tamoxifen induction has been included (Figure 3—figure supplement 2A), with the accompanying text:

“Expression of *Card14* mRNA in the skin of *Card14*^LSL-E138A/+^*Rosa26*^CreERT2/+^ mice 5d after tamoxifen injection was preferentially localized to the suprabasal layers of the epidermis (Figure 3—figure supplement 2A). […] These results suggest that keratinocytes are the main source of TNF driving skin inflammation.”

Importantly, the new *Tnfα* RNAscope data added indicate that TNF is mainly produced by skin keratinocytes in *Card14*^LSL-E138A/+^*Rosa26*^CreERT2/+^ mice following tamoxifen induction, consistent with CARD14^E138A^ signaling in keratinocytes driving skin inflammation.

Finally, we have now added text about CARD14 epidermal expression in the Discussion:

“Earlier immunohistochemical studies established that CARD14 protein was expressed in the basal layer in human skin (Jordan et al., 2012b). […] Nevertheless, regardless of apparent differences in CARD14 distribution between mouse and human, the predominant localization of CARD14 in the skin is consistent with a function in sensing harmful stimuli, such as pathogens, at the interface between the organism and the environment.”

3) Throughout the manuscript, the authors link their results to 'psoriasis' (e.g. disease markers, therapeutic response). However, the investigated CARD14 mutation is related to GPP, not (or to a lesser extent) to other psoriasis endotypes like PV (Berki et al. JID 2015). The characterization of the inflammatory response includes several markers (e.g. KI67, S100A9, cytokines and chemokines). But most are not considered specific for psoriasis, and are related to skin inflammation or an activated epidermis in general. Therefore, a more comprehensive analysis including more specific markers for psoriasiform inflammation would be preferable. It would be also ideal to compare these findings with data on these markers in human psoriasis, and comparing GPP vs. PV. This would allow the discrimination of the model phenotype and specification of the model being a true GPP or PV model. For example, RNAseq analysis would allow for such a comparison given the publicly available RNAseq data sets from psoriasis, atopic dermatitis and other inflammatory skin conditions.

To answer this important point, we have included new data (Figure 5—figure supplement 1, Figure 5—figure supplement 2 and Figure 5—figure supplement 3). The following subsection was added to the Results section to describe these figures:

“Comparison of the CARD14^E138A^ skin transcriptome with human psoriatic skin transcriptome”.

We have also added a discussion of these comparisons in the Discussion section:

“The skin transcriptome of *Card14*^LSL-E138A/+^*Rosa26*^CreERT2^ mice also shared similarities with the skin transcriptome of the GPP patient expressing the *CARD14*^E138A^ variant. The increased expression of *Steap1* and *Steap4* mRNAs in the skin of *Card14*^LSL-E138A/+^*Rosa26*^CreERT2^ mice after tamoxifen injection is particularly interesting since these genes have been found to be upregulated in several types of pustular psoriasis and are known to be important in neutrophil chemotaxis (Liang et al., 2017b).”

4) The authors assess the contribution of adaptive immunity to the phenotypes described by crossing Card14 mutants to Rag1^-/-^ mice. As no significant differences are detected in the absence of T cells and B cells, the authors conclude that adaptive immunity was not required for acute skin inflammation. This is in keeping with the relatively short time of evaluation after the induction of expression of the mutant Card14 (5 days). Is adaptive immunity relevant at later stages of the model. If so, is anti-TNF still therapeutically effective?

To address this question, we have included new experiments which investigate the importance of the adaptive immune system for skin inflammation induced by chronic CARD14^E138A^ signaling (Figure 4—figure supplement 1).

The following text was added at the Results section to describe these new data:

“To investigate the role of the adaptive immune system in skin pathology induced by prolonged CARD14^E138A^ signaling, *Card14*^LSL-E138A/+^*Rosa26*^CreERT2/+^*Rag1*^-/-^ mice were analyzed 1m after tamoxifen injection. […] Thus, the adaptive immune system was not required for skin inflammation induced by chronic CARD14^E138A^ signaling.”

The following text was added at the Discussion section:

“The absence of a requirement for adaptive immune cells for the development of skin inflammation in *Card14*^LSL-E138A/+^*Rosa26*^CreERT2/+^ mice contrasts with a published study by Wang et al. where the inflammatory skin phenotype in mice constitutively expressing CARD14^∆Q136^ was reduced by RAG1 deficiency. […] Alternatively, the inflammatory skin phenotype that develops in C*ard14*^∆Q136/+^ mice may involve developmental alterations that occur in utero that require adaptive immune cells.”

Unfortunately, we have not been able to investigate the role of TNF at later time points after tamoxifen injection due to the lockdown closure of the Crick laboratories. We plan to investigate whether anti-TNF blocks the inflammatory effects of chronic CARD14^E138A^ signaling when the Institute reopens.

5) The study included the analysis of different cytokines (Figures 1 and 2). The results raise the question as to whether ubiquitous or KC-specific expression of Card14^E138A^ result in a different inflammatory profile. A table listing all the cytokines tested and their outcome in both setting would be beneficial to easily compare and contrast between the two models. Along the same line, it would be ideal to include measurements on the cellular infiltrate in the KC-specific model.

We thank the reviewers for this suggestion. Unfortunately, we have not performed an RNAseq analysis of the skins of the *Card14*^E138A/+^*Krt14*^CreERT2^ mice and only checked the expression of selected proinflammatory transcripts relevant in skin inflammation by qRT-PCR. Nevertheless, we were able to show that keratinocyte-specific expression of CARD14^E138A^ upregulated the expression of the same genes induced by ubiquitous CARD14^E138A^ expression.

We have included Figure 2—figure supplement 2 comparing the fold changes of the assessed gene products.

The following text was added in the Results section:

“Some *Card14*^E138A/+^*Krt14*^CreERT2/+^ mice presented a visible skin phenotype without tamoxifen injection, which could be due to leaky expression of Cre allowing basal expression of CARD14^E138A^, these were excluded from experiments. […] Nevertheless, these experiments showed that CARD14^E138A^ signalling in keratinocytes alone was sufficient to induce skin inflammation.”

And we discussed the results in the following paragraph of the Discussion section:

“The fold changes of the majority of proinflammatory genes analyzed at d5 in *Card14*^LSL-E138A/+^*Krt14*^CreERT2/+^ with respect to controls were lower than the fold changes of the *Card14*^LSL-E138A/+^*Rosa26*^CreERT2/+^ mice. […] Regardless, the induced genes were the same in both Cre drivers and the phenotype of both het and hom mice in the *Krt14*^CreERT2/+^ background was similar to the *Rosa26*^CreERT2/+^ histologically and systemically demonstrating that CARD14^E138A^ signaling in keratinocytes alone is sufficient to induce skin and systemic inflammation.”

Unfortunately, we have not been able to perform experiments to characterize the immune cell infiltrate in the *Card14*^LSL-E138A/+^*Krt14*^CreERT2/+^ background due to closure of the Crick laboratories during the pandemic.

6) There are a number of experiments in which controls are missing. For example, include control Card14^+/+^ Rosa26^CreERT2/+^ in Figure 3C, Figure 3—figure supplement 1B.

We thank the reviewers for pointing out this omission. We have now included controls in all the figure panels.

7) The data in Figure 6—figure supplement 2D is relevant and would be ideal to be included in a main figure. We also suggest to include this in the main figure and replace the pictures shown in Figure 6C.

We have moved the data as suggested.

8) The finding that systemic effects are mediated through keratinocyte-driven severe skin inflammation is highly interesting and could have clinical impact. This finding deserves to be better highlighted in the manuscript. This may also suggest that patients could benefit, early in disease progression, from local anti-inflammatory therapies specifically targeting the keratinocytes, rather than systemic biologicals.

We thank the reviewers for point out this important point. We have now modified the text to highlight the potential translational impact of our study:

Abstract:

“These results suggest that anti-inflammatory therapies specifically targeting keratinocytes, rather than systemic biologicals, might be effective for GPP treatment early in disease progression.”

Introduction:

“This severe systemic phenotype resulted from CARD14^E138A^ signaling in keratinocytes and showed striking similarities to an acute flare of GPP in human patients. […] Our findings suggest that specific blockade of the CARD14 signaling pathway in keratinocytes would inhibit the systemic inflammatory syndrome that can develop in GPP patients with activating CARD14 mutations”

Discussion:

“The present study indicates that constitutive CARD14^E138A^ activation of NF-кB in keratinocytes is sufficient to promote skin inflammation and systemic inflammatory disease. This raises the possibility that it may be feasible to develop topical medicines that block CARD14 signaling in the skin to treat CARD14-induced psoriasis and systemic inflammation.”

Revisions expected in follow-up work:1) We recognize that a more detailed characterization of the model and its comparison with data sets on human psoriasis, GPP and PV would require significant additional experiments and can be performed as a follow up work. A discussion on how such an approach can complement the current findings to discriminate this with other disease models would be ideal.

We have been able to address this point with new bioinformatic analyses, comparing human skin disease and mouse transcriptomics. The results are included in the article in Figure 5—figure supplements 1, 2 and 3 and described in the subsection “Comparison of the CARD14^E138A^ skin transcriptome with human psoriatic skin transcriptome”.

2) If the authors have not tested the efficacy of anti-TNF beyond the acute period, they should clarify that this would be important to be tested in follow up work.

We have not been able to investigate the effect of anti-TNF on skin inflammation induced by chronic CARD14^E138A^ signaling due to the coronavirus pandemic. However, we plan to perform experiments to address this question in the future. As suggested by reviewers, we have included the following text:

“It will be important to test whether anti-TNF treatment can prevent chronic skin inflammation induced by CARD14^E138A^.”

3) We recognize that a thorough comparative analysis on CARD14 expression between mouse and human samples would require significant additional experiments and can be performed as a follow up work. A discussion on the implications of differential expression is warranted for this report.

We have analysed publicly available RNAseq datasets from human keratinocytes and organotypic cultures under differentiation conditions (Figure 1—figure supplement 1G). These analyses have shown that *CARD14* mRNA levels increase during human keratinocyte differentiation. We have tested several commercial antibodies against CARD14 in human skin but all of these only produced nonspecific staining. Development of better CARD14 antibodies is required to definitively determine where CARD14 is expressed in the human epidermis.

We have introduced a paragraph in the Discussion in response to the reviewers’ suggestion:

**“**Earlier immunohistochemical studies established that CARD14 protein is expressed in the basal layer in human skin (Jordan et al., 2012b). […] Nevertheless, regardless of apparent differences in CARD14 distribution between mouse and human, the predominant localization of CARD14 in the skin is consistent with a function in sensing harmful stimuli, such as pathogens, at the interface between the organism and the environment.”